# The Mutationathon highlights the importance of reaching standardization in estimates of pedigree-based germline mutation rates

Lucie A Bergeron[1]*, Søren Besenbacher[2], Tychele Turner[3], Cyril J Versoza[4], Richard J Wang[5], Alivia Lee Price[1], Ellie Armstrong[6], Meritxell Riera[7], Jedidiah Carlson[8], Hwei-yen Chen[1], Matthew W Hahn[5], Kelley Harris[8], April Snøfrid Kleppe[2], Elora H López-Nandam[9], Priya Moorjani[10], Susanne P Pfeifer[11], George P Tiley[12], Anne D Yoder[12], Guojie Zhang[1], Mikkel H Schierup[7]*

[1]Section for Ecology and Evolution, Department of Biology, University of Copenhagen, Copenhagen, Denmark; [2]Department of Molecular Medicine, Aarhus University, Aarhus, Denmark; [3]Department of Genetics, Washington University School of Medicine, St. Louis, United States; [4]Center for Evolution and Medicine, School of Life Sciences, Arizona State University, Tempe, United States; [5]Department of Biology and Department of Computer Science, Indiana University, Bloomington, United States; [6]Department of Biology, Stanford University, Stanford, United States; [7]Bioinformatics Research Centre, Aarhus University, Aarhus, Denmark; [8]Department of Genome Sciences, University of Washington, Computational Biology Division, Fred Hutchinson Cancer Research Center, Seattle, United States; [9]California Academy of Sciences, San Francisco, United States; [10]Department of Molecular and Cell Biology, Center for Computational Biology, University of California, Berkeley, Berkeley, United States; [11]Center for Evolution and Medicine, Center for Mechanisms of Evolution, School of Life Sciences, Arizona State University, Tempe, United States; [12]Department of Biology, Duke University, Durham, United States

*For correspondence:
lucie.a.bergeron@gmail.com
(LAB);
mheide@birc.au.dk (MHS)

**Competing interest:** The authors declare that no competing interests exist.

**Abstract** In the past decade, several studies have estimated the human per-generation germline mutation rate using large pedigrees. More recently, estimates for various nonhuman species have been published. However, methodological differences among studies in detecting germline mutations and estimating mutation rates make direct comparisons difficult. Here, we describe the many different steps involved in estimating pedigree-based mutation rates, including sampling, sequencing, mapping, variant calling, filtering, and appropriately accounting for false-positive and false-negative rates. For each step, we review the different methods and parameter choices that have been used in the recent literature. Additionally, we present the results from a 'Mutationathon,' a competition organized among five research labs to compare germline mutation rate estimates for a single pedigree of rhesus macaques. We report almost a twofold variation in the final estimated rate among groups using different post-alignment processing, calling, and filtering criteria, and provide details into the sources of variation across studies. Though the difference among estimates is not statistically significant, this discrepancy emphasizes the need for standardized methods in mutation rate estimations and the difficulty in comparing rates from different studies. Finally, this work aims to provide guidelines for computational and statistical benchmarks for future studies interested in identifying germline mutations from pedigrees.

## Editor's evaluation

Bergeron et al. show that mutation rate independently estimated by several teams with the same pedigree dataset can be different due to the methods and approaches used to identify de novo mutations. This result is of primary importance because it shows the necessity to have standard mutation identification methods and the difficulties to compare mutation rates from different studies.

## Introduction

Germline mutations are the source of most genetic diseases and provide the raw material for evolution. Thus, it is crucial to accurately estimate the frequency at which mutations occur in order to better understand the course of evolutionary events. The development of high-throughput next-generation sequencing offers the opportunity to directly estimate the germline mutation rate over a single generation, based on a whole-genome comparison of pedigree samples (mother, father, and offspring), without requiring assumptions about generation times or fossil calibrations (*Tiley et al., 2020*). Pedigree sequencing provides multiple pieces of information in addition to an overall mutation rate. For instance, the genomic locations, the spectrum of mutation types (e.g., transition or transversion), and the nucleotide context of all mutations can easily be gleaned. Furthermore, pedigree sequencing enables researchers to identify the parental origin of the mutations; that is, whether the mutation arose in the maternal or paternal germline. Finally, using pedigrees means that researchers often have precise information about the age of the parents at the time of reproduction, and comparing several trios (i.e., three related individuals: mother, father, and offspring) at different parental ages can tell us about the effect of parental age on the total number of transmitted mutations, their location, and their spectrum. Thus, there has been a growing interest in applying this method to address medical and evolutionary questions.

The first estimate of the human germline mutation rate using pedigrees was published more than 10 years ago (*Roach et al., 2010*). Four years later, the first pedigree-based mutation rate for a nonhuman primate, the chimpanzee, was estimated (*Venn et al., 2014*). Today, at least 20 vertebrate species have mutation rates estimated by pedigree sequencing (*Table 1*), with half added in the past two years. Each study differs in the number of trios, the sequencing technology and depth, the ages of individuals included, and the bioinformatics pipelines used to analyze the data (see *Table 1* and *Supplementary file 1a*). Thus, reported variation in mutation rates among studies might result from a combination of biological and methodological factors. Although most studies using human pedigrees have now reached similar rates of ~$1.2 \times 10^{-8}$ mutations per site per generation at an average age of around 30 years (*Table 1*), the effect of different methodologies is likely to have a much larger effect on estimates in other species. This is because these species have lower-quality genome assemblies, less information about segregating polymorphisms, often higher heterozygosity, and an overall deficit in prior information on mutation rates. With an increasing number of studies being published, an examination of the differences among studies and suggestions for standards that will minimize differences caused by methodological discrepancies are warranted.

The key principle of the pedigree-based approach is to detect de novo mutations (DNMs) present in a heterozygous state in an offspring that are absent from its parents' genomes (*Figure 1*). A per-site per-generation mutation rate can be inferred by dividing the number of DNMs by the number of sites in the genome that mutations could possibly be identified in (and accounting for the diploid length of the genome, as mutations can be transmitted by both the mother and the father). As mutations are rare events, detecting all the true DNMs (or having a high sensitivity) while avoiding errors (or increasing precision) from a single generation remains challenging. False-positive (FP) calls (sites incorrectly detected as DNMs) can be caused by sequencing errors, errors introduced by read mapping and genotyping steps, stochastically missing an alternative allele in a parent, or somatic mutations in the offspring. Numerous filters are thus often applied on the variant sites to increase the precision of the candidate DNMs' detection. However, filters that are too conservative can also discard true DNMs, reducing the sensitivity by increasing the rate of false-negative calls (true DNMs not detected). Therefore, a balance should be found between precision and sensitivity – a goal that has led to the development of multiple different methods to estimate germline mutation rates from pedigree samples.

**Table 1.** Vertebrate species with a direct estimate of the mutation rate using a pedigree approach. The list of species includes 10 primates, 5 nonprimate mammals, 1 bird, and 4 fish (see *Supplementary file 1b* for differences in study design and methodology).

| Species | Mutation rate per site per generation: $\mu \times 10^{-8}$ | Number of trios | Parental age* | Reference |
|---|---|---|---|---|
| Orangutan (*Pongo abelii*) | 1.66 | 1 | ♂: 31.00 and ♀: 15.00 | *Besenbacher et al., 2019* |
| Human (*Homo sapiens*) | 1.17<br>0.97<br>1.20<br>1.20<br>1.28<br>1.05<br>1.29<br>1.28<br>1.30<br>1.10<br>1.22 | 1 (CEU)<br>1 (YRI)<br>78<br>269<br>13<br>719<br>1550<br>150<br>516<br>593<br>1449 | Unspecified<br>Unspecified<br>♂: 29.10 and ♀: 26.50<br>Unspecified<br>♂: 29.80<br>♂: 33.40<br>Unspecified<br>~27.70<br>♂: 33.40<br>♂: 29.10 and ♀: 26.00<br>♂: 29.70 and ♀: 26.90 | *Conrad et al., 2011*<br>*Conrad et al., 2011*<br>*Kong et al., 2012*<br>*Francioli et al., 2015*<br>*Rahbari et al., 2016*<br>*Wong et al., 2016*<br>*Jónsson et al., 2017*<br>*Maretty et al., 2017*<br>*Turner et al., 2017*<br>*Sasani et al., 2019*<br>*Kessler et al., 2020* |
| Chimpanzee (*Pan troglodytes*) | 1.20<br>1.48<br>1.26 | 6<br>1<br>7 | ♂: 18.90 and ♀: 15.00<br>♂: 24.00 and ♀: 24.00<br>♂: 19.30 and ♀: 15.90 | *Venn et al., 2014*<br>*Tatsumoto et al., 2017*<br>*Besenbacher et al., 2019* |
| Gorilla (*Gorilla gorilla*) | 1.13 | 2 | ♂: 14.50 and ♀: 20.50 | *Besenbacher et al., 2019* |
| Baboon (*Papio anubis*) | 0.57 | 12 | ♂: 10.70 and ♀: 10.20 | *Wu et al., 2020* |
| Rhesus macaque (*Macaca mulatta*) | 0.58<br>0.77 | 14<br>19 | ♂: 7.80 and ♀: 7.10<br>♂: 12.40 and ♀: 8.40 | *Wang et al., 2020*<br>*Bergeron et al., 2021a* |
| Green monkey (*Chlorocebus sabaeus*) | 0.94 | 3 | ♂: 8.70 and ♀: 4.70 | *Pfeifer, 2017* |
| Owl monkey (*Aotus nancymaae*) | 0.81 | 14 | ♂: 6.60 and ♀: 6.50 | *Thomas et al., 2018* |
| Marmoset (*Callithrix jacchus*) | 0.43 | 1 | ~2.80 | *Yang et al., 2021* |
| Gray mouse lemur (*Microcebus murinus*) | 1.52 | 2 | ♂: 4.55 and ♀: 1.45 | *Campbell et al., 2021* |
| Mouse (*Mus musculus*) | 0.57<br>0.39 | 8<br>15 | Unspecified<br>~0.47 | *Milholland et al., 2017*<br>*Lindsay et al., 2019* |
| Cattle (*Bos taurus*) | 1.17 | 5 | Unspecified | *Harland et al., 2017* |
| Wolf (*Canis lupus*) | 0.45 | 4 | ♂: 4.00 and ♀: 2.25 | *Koch et al., 2019* |
| Domestic cat (*Felis catus*) | 0.86 | 11 | ♂: 4.70 and ♀: 2.90 | *Wang et al., 2021b* |
| Platypus (*Ornithorhynchus anatinus*) | 0.70 | 2 | Unspecified | *Martin et al., 2018* |
| Collared flycatcher (*Ficedula albicollis*) | 0.46 | 7 | Unspecified | *Smeds et al., 2016* |
| Herring (*Clupea harengus*) | 0.20 | 12 | Unspecified | *Feng et al., 2017* |

*Table 1 continued on next page*

*Table 1 continued*

| Species | Mutation rate per site per generation: $\mu \times 10^{-8}$ | Number of trios | Parental age* | Reference |
| --- | --- | --- | --- | --- |
| Cichlid (*Astatotilapia calliptera, Aulonocara stuartgranti*, and *Lethrinops lethrinus*) | 0.35 | 9 | Unspecified | *Malinsky et al., 2018* |

*Depending on the study, the parental ages are reported as average paternal age (♂), average maternal age (♀), average parental age (~), or unspecified.

In this study, we aim to define what we consider to be the state of the art in pedigree-based germline mutation rate estimation, to discuss the pros and cons of each methodological step, and to summarize best practices that should be used when calling germline mutations. We review several recently published methods that estimate germline mutation rates from pedigree samples. In parallel, we set up a competition – the 'Mutationathon' – among five research groups to explore the effect of different methodologies on mutation rate estimates. Using a common genomic dataset consisting of a pedigree of the rhesus macaque (*Macaca mulatta*; *Bergeron et al., 2021a*), each group estimated the number of candidate DNMs (validated by PCR amplification and Sanger resequencing) and a germline mutation rate. An examination of the estimated rates produced by different groups not only highlighted the choices that can be made in estimating per-generation mutation rates, but it also provided us with an opportunity to characterize the impact of these choices on the systematic differences in estimated rates, which in turn yielded important insights into the parameters that could reduce the occurrence of FP calls.

# Results and discussion
## Comparison of methods

The overall pipeline from high-throughput next-generation sequencing data to an estimated mutation rate is similar across all studies listed in *Table 1*. It includes five steps (*Figure 2*):

1. sampling and whole-genome sequencing of at least one trio or extended pedigrees that also include a third generation (useful for validation of putative DNMs in the offspring),
2. alignment of reads to a reference genome and post-processing of alignments,
3. variant calling to infer genotypes or genotype likelihoods for all individuals,

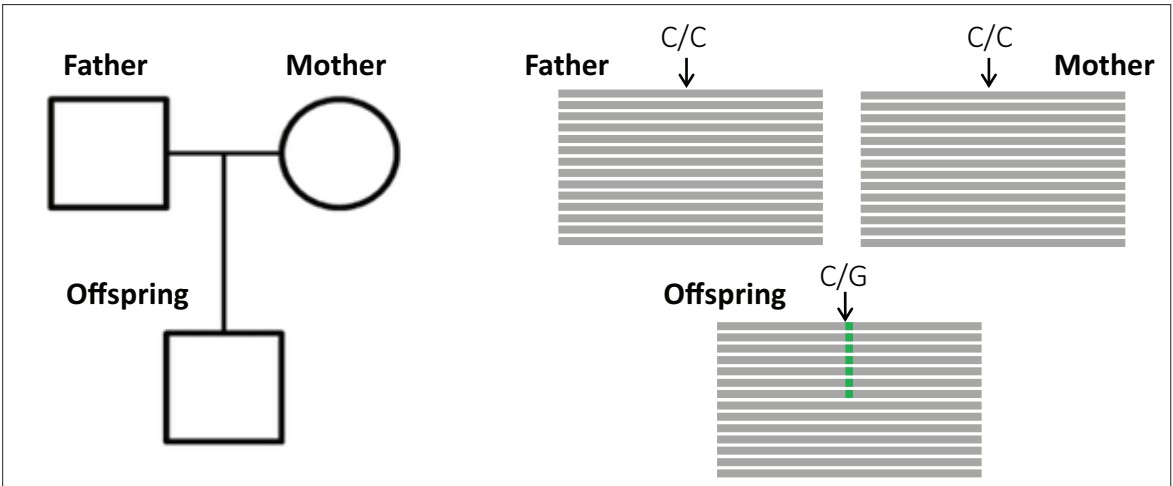

**Figure 1.** Detection of a de novo mutation (DNM) in a trio sample (mother, father, and offspring). Potential candidates for DNMs are sites where approximately half of the reads (indicated as gray bars) from the offspring have a variant (indicated in green) that is absent from the parental reads.

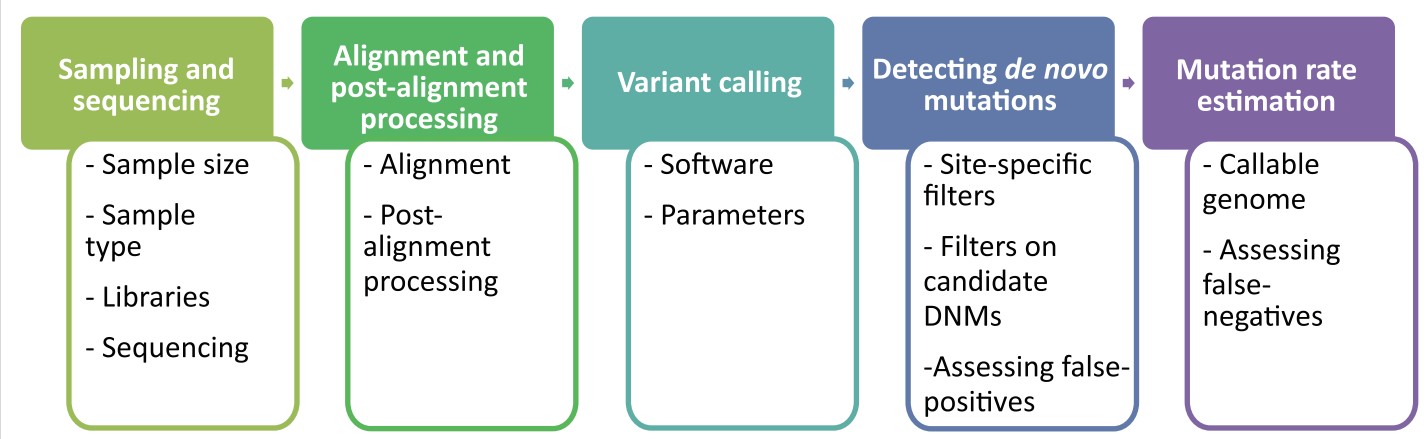

**Figure 2.** Flow of the main steps to call de novo mutations (DNMs) from pedigree samples. Each step lists the various choices in study design and methodology that might impact mutation rate estimates.

4. detection of DNMs via filtering of candidates (including an assessment of the false discovery rate [FDR]), and finally
5. the estimation of a per-generation mutation rate accounting for the length of the accessible genome (including an assessment of the false-negative rate [FNR]).

## Step 1: Sampling and sequencing

### Sample size

Pedigree-based study designs can vary significantly, from those that include only one trio per species (e.g., *Besenbacher et al., 2019*) to those that include thousands of trios (e.g., *Halldorsson et al., 2019*). The first study to estimate a pedigree-based human mutation rate used only two trios and estimated a mutation rate of $1.1 \times 10^{-8}$ per site per generation (*Roach et al., 2010*), which is within the overall variation reported across studies with larger sample sizes (*Table 1*). Larger sample sizes reduce uncertainty in the average mutation rate for a species and offer more statistical power for the exploration of various parameters such as the parental age effect, the contribution of each parent to the total number of DNMs, and the distribution of mutations across genomes. Multi-sibling pedigrees (i.e., when there is more than one offspring) offer a unique opportunity to detect mutations that may be mosaic within one of the parents indicative of having occurred early in development. Indeed, if, for instance, a paternal DNM is detected in more than one sibling from a pedigree, it is unlikely that the same mutation occurred in different sperm cells. Instead, an early postzygotic mutation may have occurred in primordial germ cells (PGCs) during embryonic development of the father. Therefore, the mutation would be absent from the father's somatic tissue, while affecting more than one of his descendants. Moreover, by means of haplotype sharing with a third noncarrier sibling, DNMs that arose before the PGC specification can be detected, even if present in the parental somatic tissue sampled (e.g., *Jónsson et al., 2018*). Multigeneration pedigrees, also referred to as extended trios, can be used to validate true DNMs and to adjust quality filters by studying transmission to a third generation. Multigeneration pedigrees also allow researchers to easily determine whether these transmitted mutations came from the maternal or paternal parent in the first generation (e.g., *Jónsson et al., 2017*). Therefore, whenever possible, multiple trios should be analyzed and more than two generations should be included. Finally, the age of the parents at the time of reproduction is required for estimating the per-year mutation rate from the per-generation rates directly measured in the trios. In some studies, the age of the parents at conception is not available, and instead, the mean age of reproduction is used for the estimation of the per-year mutation rate. While useful, this approximation can lead to biased results if the age of the parents at conception was much older or much younger compared to the mean age in the population. Thus, when possible, the information on the age of each parent at the time of conception should be collected as it is essential for the interpretation of results and to help understand parental age effects on mutation rate.

## Sample type

The most commonly used sample types are somatic tissues such as whole blood, muscle, or liver, which generally produce a high quantity of DNA with long fragment sizes and allow for high-coverage sequencing. The duration and temperature of storage can affect the quality of the extracted DNA and increase the rate of sequencing errors. Thus, to minimize DNA damage during storage, DNA is typically kept in TE (tris-EDTA) buffer. Moreover, it is advised to store DNA at –80°C for long-term storage (months to years) and in liquid nitrogen at –164°C for decades (*Baust, 2008*; *Straube and Juen, 2013*). Other materials such as buccal swabs or fur can be considered, but they can be technically challenging. For instance, as part of a recent study on rhesus macaques (*Bergeron et al., 2021a*), DNA was extracted from hair samples and sequenced at 95× coverage, yet, due to the fragmentation, only 38% of the reads were mappable to the reference genome. After variant calling, the average depth of usable reads was 6×, with only 10% of sites covered by more than 10 reads. To reduce the number of FP calls caused by somatic mutations, it is best to avoid tissues with an accumulation of such mutations, such as skin. In this regard, blood is often the preferred tissue: as many different tissues contribute cells to the blood, the hope is that a somatic mutation in any one of them will not be mistaken for a DNM. However, in rare cases, mainly in older individuals, clonal hematopoiesis can lead to high-frequency somatic mutations in the blood. Thus, sequencing more than one type of tissue, when feasible, should be considered. Comparing the DNMs called from different tissues could reduce the potential for mistaking somatic mutations as DNMs. If only one tissue is available, allelic balance of both candidate DNMs and known single-nucleotide polymorphisms (SNPs) should allow for better detection of somatic mutations.

## Libraries

After DNA extraction, genomic library preparation is another step that can introduce sequencing errors. Most studies have used Illumina sequencing platforms, yet, even for a single technology, there are different library preparation protocols available. PCR amplification is commonly used to increase the quantity of DNA, but this can generate artifacts caused by the introduction of sequence errors (PCR errors) or by the overamplification of some reads (PCR bias) (*Acinas et al., 2005*). Thus, for samples yielding a sufficient amount of DNA, PCR-free libraries that do not involve amplification prior to cluster generation are preferable. Moreover, as different library preparation methods can result in different amplification biases (*Ross et al., 2013*; *Wingett, 2017*), utilizing different types of library preparations may be advisable to reduce the sources of error.

## Sequencing

All Illumina sequencing platforms use similar sequencing chemistry (sequencing-by-synthesis) and mainly differ in running speed and throughput. Another equivalent technology, used in two studies (*Bergeron et al., 2021a*; *Roach et al., 2010*), is BGISEQ-500, combining DNA nanoball nanoarrays with polymerase-based stepwise sequencing (*Mak et al., 2017*) and showing similar performances to Illumina on data quality (*Chen et al., 2019*; *Patch et al., 2018*). Another study used 10X Genomics-linked reads, which can help phase maternal and paternal mutations (*Campbell et al., 2021*). However, it remains unclear if alternative library preparation and sequencing platforms introduce additional biases compared to standard Illumina protocols. Most pedigree-based studies of germline mutations have sequenced each individual to a depth between 30× and 50× (*Besenbacher et al., 2019*; *Campbell et al., 2021*; *Jónsson et al., 2017*; *Kessler et al., 2020*; *Malinsky et al., 2018*; *Milholland et al., 2017*; *Sasani et al., 2019*; *Smeds et al., 2016*; *Thomas et al., 2018*; *Turner et al., 2017*; *Wang et al., 2020*; *Wu et al., 2020*), three studies sequenced at a higher depth of 80× (*Bergeron et al., 2021a*; *Maretty et al., 2017*) and 150× (*Tatsumoto et al., 2017*), while six studies sequenced at a depth lower than 25× on average (*Harland et al., 2017*; *Koch et al., 2019*; *Lindsay et al., 2019*; *Martin et al., 2018*; *Pfeifer, 2017*; *Rahbari et al., 2016*). A minimum coverage of 15× has been advised to call SNPs accurately (*Fumagalli et al., 2013*). Yet, this depth might not be sufficient to call germline mutations since it might be hard to distinguish genuine germline mutations from somatic mutations that are present in a substantial fraction of cells. Furthermore, with low coverage the probability of calling a parent homozygous for the reference allele, when they are actually heterozygous, becomes non-negligible at the genome-wide level. For example, the binomial probability of not observing a read with one of the alleles in a heterozygote with 15× coverage is $0.5^{15} = 3.05 \times 10^{-5}$, which will

happen by chance around 30× in a genome with 1 million heterozygous positions. Likewise, based on the binomial distribution, the probability that a somatic mutation present in 10% of cells is seen in more than 30% of reads is 0.0113 with 20× coverage but falls to 0.0004 with 35× coverage. Thus, it is advised to aim for a minimum of 35× as a rule of thumb.

## Step 2: Alignment and post-alignment processing

### Alignment

To find DNMs, we must first find where in the genome each of the short sequencing reads comes from. The Burrows–Wheeler Aligner (BWA; *Li and Durbin, 2009*) is an algorithm developed to map short reads (50–250 bp) to a reference genome and has been used in the majority of studies on direct mutation rate estimation (*Bergeron et al., 2021a*; *Besenbacher et al., 2019*; *Harland et al., 2017*; *Jónsson et al., 2017*; *Kessler et al., 2020*; *Koch et al., 2019*; *Malinsky et al., 2018*; *Maretty et al., 2017*; *Milholland et al., 2017*; *Pfeifer, 2017*; *Sasani et al., 2019*; *Smeds et al., 2016*; *Tatsumoto et al., 2017*; *Thomas et al., 2018*; *Turner et al., 2017*; *Wang et al., 2020*; *Wu et al., 2020*). In particular, the BWA-MEM algorithm is fast, accurate, and can be implemented with an insert size option to improve the matching of paired reads. Several aspects of the study organism and study design can have detrimental effects on read mapping. Some studies reported a trimming step to remove adapter sequences and poor-quality reads – those with a high proportion of unknown ('N') bases or low-quality-score bases (*Bergeron et al., 2021a*; *Maretty et al., 2017*; *Tatsumoto et al., 2017*; *Wu et al., 2020*). However, trimming might not be necessary as some mapping software will soft-clip (or mask) the adaptors, while low-quality reads can be removed during the variant-calling step. The quality of the reference genome can play an important role in obtaining a large proportion of reads with high mapping scores. In the case of a poor or nonexistent reference genome, using the reference genome of a phylogenetically related species is an option, but this could make the downstream analysis more complex (*Prasad and Lorenzen, 2021*). Moreover, BWA was designed to map low-divergence sequences, so that using a related species, or even a closely related individual in the same species when heterozygosity is high, could impact the mapping. Finally, low-complexity regions (LCRs) and repetitive sequences such as dinucleotide tandem repeats can be problematic for read mapping. Standards have been proposed for human genome analysis and can be followed for germline mutation rate calling in species with comparable heterozygosity (for details, see *Regier et al., 2018*).

### Post-alignment processing

To correct for possible misalignment of sequencing reads to the reference genome, post-alignment quality control is necessary. This step often includes base quality score recalibration (BQSR), removing of duplicate reads, and realignment around indels. BQSR corrects for any bias in the base quality score assigned by the sequencer by utilizing information from a set of known variants for the studied species. When such a dataset is not available, as in many nonhuman species, the Best Practices of the Genome Analysis ToolKit (GATK) software from the Broad Institute advises to proceed first with variant calling in all available samples and subsequently using the best-quality variants to recalibrate the base quality scores (*GATK team, 2021*). If multiple generations are available, high-quality variants fully transmitted across generations can be used for BQSR (*Wu et al., 2020*). However, some studies have ignored this step due to the circularity of this method and its computational expense, as variants will be called twice (*Bergeron et al., 2021a*; *Thomas et al., 2018*; *Wang et al., 2020*). A comparative study presented a difference of less than 0.1% between the total variant sites called with and without recalibration (*Li, 2014*), and this difference was even lower for high-coverage (40×) sequencing (*Tian et al., 2016*); however, this step is still advised to increase the quality of variant calling (*Li, 2020*). Duplicates, identical reads due to amplification (PCR duplicates) or sequencing clusters (optical duplicates), can increase FP calls and erroneously inflate sequencing coverage. Therefore, duplicates should be marked or removed even for sequences from PCR-free libraries. Reads terminating with indels are more likely to be misaligned; thus, depending on the variant caller used, realignment around indels may be advised to correct for this artifact. Specifically, realignment around indels is required when calling variants with non-haplotype-aware callers (such as GATK's UnifiedGenotyper), but is not necessary with haplotype-aware variant callers (such as GATK HaplotypeCaller [*Poplin et al., 2018*], Platypus [*Rimmer et al., 2014*], or FreeBayes [*Garrison and Marth, 2012*]). From GATK release 3.6 onward, the realigned reads around indels can be outputted during the variant-calling step. Alternatively, BWA

alignments can be used to construct a variation-aware graph with GraphTyper (*Eggertsson et al., 2017*), including known polymorphisms and newly genotyped variants. Thereby, reads are realigned to the graph, reducing reference bias and improving read alignment near indels (*Eggertsson et al., 2017*) and structural variants (*Eggertsson et al., 2019*). Finally, other quality controls can be applied after mapping, such as removing reads mapping to multiple locations, as they could map with a good mapping quality in two or more locations and be ignored by further quality filters. However, the overall impact of many of these filters, such as BQSR and realignment around indels, on the final set of DNMs has not yet been studied.

## Step 3: Variant calling
### Software
Different algorithms have been shown to perform similarly well in calling nucleotide variants (*Li, 2014*). GATK (*Van der Auwera and O'Connor, 2020*) is widely used among studies that call germline DNMs (*Bergeron et al., 2021a*; *Besenbacher et al., 2019*; *Campbell et al., 2021*; *Feng et al., 2017*; *Harland et al., 2017*; *Jónsson et al., 2017*; *Koch et al., 2019*; *Malinsky et al., 2018*; *Maretty et al., 2017*; *Milholland et al., 2017*; *Pfeifer, 2017*; *Sasani et al., 2019*; *Smeds et al., 2016*; *Tatsumoto et al., 2017*; *Thomas et al., 2018*; *Turner et al., 2017*; *Wang et al., 2020*; *Wong et al., 2016*; *Wu et al., 2020*). Other commonly used variant callers are GraphTyper (*Eggertsson et al., 2017*; e.g., utilized by *Beyter et al., 2021*; *Halldorsson et al., 2019*; *Jonsson et al., 2021*; *Jónsson et al., 2018*) and FreeBayes (*Garrison and Marth, 2012*; e.g., utilized by *Turner et al., 2017*). Using more than one variant caller can increase confidence in the SNP set but can become computationally expensive (*Turner et al., 2017*).

### Parameters
Even within the same variant caller, different methods can be used (see *Supplementary file 1a*). For instance, in GATK v3, three strategies are available: (1) per-sample variant calling; (2) batch calling, in which samples are analyzed separately and concatenated for downstream analysis; and (3) joint calling, in which variants are called simultaneously across all samples (with the UnifiedGenotyper command). In GATK v4, the new recommendation is to first call variants for each sample separately (Haplotype-Caller in ERC mode), and then combine all the samples (GenomicsDBImport) to jointly genotype them (GenotypeGVCFs); thus, the initial identification of variant sites is separable from the assignment of genotypes to each individual. UnifiedGenotyper and HaplotypeCaller should have a similar ability to detect SNPs, but differences in variant sets have been observed (*Lescai et al., 2014*). Moreover, the GATK HaplotypeCaller ERC mode has two options: the BP_RESOLUTION option provides records for every single site in the genome, even nonvariant sites, while the GVCF option groups the nonvariant sites into a block of record. This variant-calling step is computationally expensive, especially if variants are called in BP-RESOLUTION mode, but it can be useful to determine the part of the genome in which there is full power to detect mutations. It is still unclear which strategy should be prioritized; thus, it is advised to report the method used and any additional options that have been implemented. The default settings of GATK applied during variant calling should also be kept in mind. For instance, the heterozygosity prior is by default at 0.001, which could have an impact when analyzing species with much higher or much lower heterozygosity, though the effect of this prior has not been evaluated in the context of mutation rate studies.

## Step 4: Detecting de novo mutations
### Site-specific filters
Variant information is stored in the 'variant-calling file (.vcf)' file format, which includes different types of information on the quality of the genotype calls (see *Box 1*). Thus, the first set of filters (i.e., site-specific filters) can be applied to ensure that there is a true variant at a particular position. GATK's Best Practices (*Van der Auwera and O'Connor, 2020*) advise to perform a Variant Quality Score Recalibration (VQSR) step to ensure that genotypes are correctly called. However, this tool is not suitable for DNMs as it would remove many rare variants; instead, hard filtering should be applied. GATK provides some general recommendations for these site-specific filters, warning that these should be a starting point and filters may need to be adjusted depending on the callset or the species studied (*GATK team, 2020*).

## Box 1. The variant call format.

The variant call format, or vcf, is a text format for storing information on genetic variants. Each line in the file corresponds to a particular variant detected and provides information on:

- CHROM and POS: the position of the variant (chromosome name and site);
- REF and ALT: the reference and alternative alleles;
- QUAL: the p-Phred probability that a variant is actually present at that site;
- FILTER: after filtering, PASS indicates that the variant position passes the filtering;
- INFO: site-specific annotations;
- FORMAT: sample-specific annotations; and
- an additional column per sample with the values for the FORMAT annotations.

The sample-specific annotations provide information on the quality of a particular genotype for a sample and include:

- DP: the read depth;
- AD: the allelic depth for the reference and alternative allele; and
- GQ: the genotype quality.

On the other hand, the site-specific annotations are informative as to whether a site has an alternative allele or not and include:

- QD: quality of a call at a given site taking into account the depth (QUAL/DP);
- MQ: the root mean square of the mapping quality at the position;
- FS and SOR: information on strand bias;
- MQRankSum: the comparison of the mapping quality of reads carrying an alternative or a reference allele;
- BaseQRankSum: the comparison of the base quality of reads carrying an alternative or a reference allele; and
- ReadPosRankSum: position bias within reads.

The currently advised hard filter criteria for germline short variant discovery are QD < 2.0; MQ < 40.0; FS > 60.0; SOR > 3.0; MQRankSum < –12.5; ReadPosRankSum < –8.0 (see *Box 1* and *Supplementary file 1b* for details on each filter). Although some studies followed these best practices (*Jónsson et al., 2017*; *Wu et al., 2020*), others implemented only a subset of filters (e.g., three studies reported the GATK filters without SOR > 3.0; *Koch et al., 2019*; *Thomas et al., 2018*; *Wang et al., 2020*) and *Besenbacher et al., 2019* kept only four parameters – FS, ReadPosRankSum, BaseQualityRankSum, and MQRankSum – as they are calculated based on statistical tests following a known distribution or readjusted the filtering thresholds based on previous results (e.g., *Koch et al., 2019* changed the Read-PosRankSum threshold from –8 to 15 while *Bergeron et al., 2021a* changed the site filters by first using the advised parameters and then adjusting them to reduce the apparent FP calls). Several of the earlier studies implemented a different suite of site filters altogether (*Pfeifer, 2017*; *Smeds et al., 2016*; *Tatsumoto et al., 2017*). Given this plethora of choices, we suggest that reporting filter details should be common practice to improve the comparability of mutation rate estimates. Another site-specific filter is the Phred-scaled probability that a certain site is polymorphic in one or more individuals (QUAL), which has been used in some studies (*Harland et al., 2017*; *Pfeifer, 2017*; *Wu et al., 2020*).

## Filters of candidate DNMs

From pedigrees, germline mutations are detected as 'Mendelian violations' where at least one of the alleles observed in the offspring is absent from both of its parents. Most mutation rate studies

restrict Mendelian violations to sites where both parents are homozygous for the reference allele (HomRef; 0/0) and the offspring is heterozygous (Het; 0/1 or 1/0) (*Bergeron et al., 2021a*; *Besenbacher et al., 2019*; *Jónsson et al., 2017*; *Koch et al., 2019*; *Pfeifer, 2017*; *Smeds et al., 2016*; *Thomas et al., 2018*; *Wang et al., 2020*; *Wu et al., 2020*). Other combinations of genotypes could also be caused by germline mutations such as parents homozygous for the alternative allele (HomAlt; 1/1) with heterozygous offspring (0/1 or 1/0), or one parent HomRef (0/0) and the other HomAlt (1/1) with an offspring either HomRef (0/0) or HomAlt (1/1). These sites are usually filtered out and assumed to represent a small portion of the genome to avoid the added uncertainty associated with these genotypes (*Wang et al., 2021a*). However, before excluding these sites, researchers should note that their expected frequency increases with the level of heterozygosity of the species studied and the phylogenetic distance to the reference genome used for mapping. For a phylogenetic distance to the reference genome of 2%, ~1 in 50 true DNMs is expected to occur in a background where both parents are homozygous for the alternative allele (1/1). After selecting the final set of Mendelian violations, several sample-specific filters are applied to ensure the genotypes of each individual are of high quality and to reduce FP calls. These individual filters and thresholds used vary substantially between studies (see *Supplementary file 1c*), but generally include a depth filter (i.e., the number of reads for each individual at a particular site), a genotype quality filter (i.e., the Phred-scaled confidence of the assigned genotype), as well as a filter on the allelic depth (i.e., the number of reads supporting the alternative allele and the reference allele).

Sites with low read depth (DP) are prone to exhibit Mendelian violations due to stochastic sampling of reads and due to sequencing and genotyping errors, while positions with particularly high depth could indicate a misalignment of reads in low complexity or paralogous regions. As each study analyzed pedigrees sequenced at various depths, different cutoffs were chosen for this filter, some more permissive than others. Some studies only set a minimum DP of approximately 10 reads (e.g., *Jónsson et al., 2017*; *Pfeifer, 2017*; *Sasani et al., 2019*), while other higher-coverage studies were able to set more conservative minimum and maximum thresholds, varying from a minimum of 10–20 to a maximum of 60–150 (e.g., *Maretty et al., 2017*: DP < 10 and DP > 150; *Thomas et al., 2018*: DP < 20 and DP > 60; *Wang et al., 2020*: DP < 20 and DP > 60). Another approach is to use a relative depth threshold for each individual (e.g., depth$_{individual}$ ± 3$\sigma$, with $\sigma$ being the standard deviation around the average depth [*Tatsumoto et al., 2017*], or a maximum threshold of 2 × depth$_{individual}$ [*Besenbacher et al., 2019*]) or, when all individuals were sequenced at a similar depth, an relative depth per trio (e.g., a DP filter of 0.5 × depth$_{trio}$ and 2 × depth$_{trio}$ [*Bergeron et al., 2021a*]). Alternatively, *Rahbari et al., 2016* and *Wu et al., 2020* tested if the depth at each site followed a Poisson distribution under the null hypothesis that lambda was depth$_{individual}$, and filtered away sites where at least one individual of the trio had a p-value higher than 2 × 10$^{-4}$.

To correct for genotyping errors, two parameters from the output .vcf can be used: the Phred-scaled likelihood of the genotype (PL) and the genotype quality (GQ). The most likely genotype has a PL of 0, while the least likely genotype has the highest PL value. GQ is the difference between the PL$_{2nd\ most\ likely}$ and PL$_{1st\ most\ likely}$, with a maximum reported of 99. Applied GQ thresholds vary between 20 (*Jónsson et al., 2017*; *Sasani et al., 2019*) and 70 (*Wang et al., 2021b*, *Wang et al., 2020*). Instead of using GQ, some studies used the difference between PL$_{2nd\ most\ likely}$ and PL$_{1st\ most\ likely}$, which is not limited to a maximum of 99, and applied more conservative criteria for the offspring heterozygous genotype than for the homozygous parents (*Maretty et al., 2017*: homozygous PL$_{2nd\ most\ likely}$ − PL$_{1st\ most\ likely}$ < 80, heterozygous PL$_{2nd\ most\ likely}$ − PL$_{1st\ most\ likely}$ < 250; *Tatsumoto et al., 2017*: homozygous PL$_{2nd\ most\ likely}$ − PL$_{1st\ most\ likely}$ < 100, heterozygous PL$_{2nd\ most\ likely}$ − PL$_{1st\ most\ likely}$ < 200).

Variants can also be filtered using allelic depth: the number of reads supporting the reference allele and the alternative allele. To ensure the homozygosity of the parents, some studies filter away sites where alternative alleles are present in the parents' reads. AD refers to the number of reads supporting the alternative allele, with previously utilized thresholds include AD > 0 (*Besenbacher et al., 2019*; *Harland et al., 2017*; *Koch et al., 2019*; *Pfeifer, 2017*; *Sasani et al., 2019*; *Smeds et al., 2016*; *Wang et al., 2021b*), AD > 1 (*Jónsson et al., 2017*; *Wang et al., 2020*), or AD > 4 (*Maretty et al., 2017*). Even more conservative, one study used a lowQ AD2 > 1, that is, the number of alternative alleles in the low-quality reads (not used for variant calling) should not exceed 1 (*Besenbacher et al., 2019*).

Allelic depth is also used to calculate the allelic balance (AB): the proportion of reads supporting the alternative allele relative to the total depth at this position. In the case of a DNM, the offspring

should have approximately 50% of its reads supporting each allele. Purely somatic mutations are expected to cause only a small fraction of reads to carry an alternate allele, though this fraction can be different for mutations occurring early in the zygote stage of the offspring and leading to germline mosaicism. A previous large-scale analysis of human pedigrees recovered a bimodal allelic balance distribution of Mendelian violations in the offspring before applying an AB filter, with a peak around 50% interpreted as DNMs, and another peak around 20% likely corresponding to somatic mutations (*Besenbacher et al., 2015*), mismapping errors, or sample contamination (*Karczewski et al., 2019*). Thus, careful filtering on AB is required to avoid FPs. Thresholds used for the AB filter vary between a minimum of 20% (*Pfeifer, 2017*) to 40% (*Thomas et al., 2018*), and a maximum, when applied, of 60% (*Thomas et al., 2018*) to 75% (*Jónsson et al., 2017*). Instead of a hard cutoff, one study used a binomial test on the allelic balance under the null hypothesis of a 0.5 frequency, removing positions with a p-value lower than 0.05 (*Wu et al., 2020*).

Additional filters can be used, for instance, to remove candidate DNMs present in individuals other than the focal offspring, including siblings (*Pfeifer, 2017*; *Smeds et al., 2016*), only unrelated individuals in the same dataset (*Bergeron et al., 2021a*; *Besenbacher et al., 2019*; *Campbell et al., 2021*; *Thomas et al., 2018*; *Wu et al., 2020*), or polymorphism datasets of the same species (*Pfeifer, 2017*; *Smeds et al., 2016*; *Wu et al., 2020*). This filter is based on the idea that the chance of getting a DNM at a position already being polymorphic is very low unless there is very high heterozygosity, thus guarding against the possibility that a heterozygous site was missed in the parents. However, recurrent mutations have been reported, especially at CpG locations (*Acuna-Hidalgo et al., 2016*; *Ségurel et al., 2014*). Filters can also be applied to the distance between mutations, again assuming that the probability of having two mutations close to each other is low. For instance, in some studies, candidate DNMs were removed if four or more candidates were located in a 200 base-pairs window (*Koch et al., 2019*) or two candidates were less than 10 base-pairs (*Tatsumoto et al., 2017*) or 100 base-pairs apart (*Wu et al., 2020*) from each other. Here again, the underlying assumptions is not always fulfilled as there is evidence of nonrandom clustering of mutations (*Brandler et al., 2016*; *Turner et al., 2016*). In humans, ~3% of the DNMs are part of a mutation cluster (*Besenbacher et al., 2016*; *Kaplanis et al., 2019*), a feature that appears to be conserved among all eukaryotes (*Schrider et al., 2011*). Therefore, instead of discarding these mutations, exploring their proportion may be an additional quality filter. Finally, some studies removed DNM candidates located in the LCRs (*Sasani et al., 2019*) or repetitive regions of the genome (*Pfeifer, 2017*) that are prone to mismapping.

## Assessing FPs

After choosing each filter according to the dataset, a total number of candidate DNMs per offspring is found. Yet, as stringent as the filters can be, there are still chances for FPs to be introduced in the final set of DNMs. Even though there is no perfect method to correct the FP calls, this issue should be addressed.

One of the most straightforward methods to validate DNMs is by PCR amplification followed by resequencing such as Sanger sequencing to ensure the genotype of each individual of the trio (*Bergeron et al., 2021a*; *Koch et al., 2019*; *Maretty et al., 2017*; *Tatsumoto et al., 2017*; *Wu et al., 2020*). However, this PCR amplification and resequencing method can be challenging. In addition to the cost, designing primers for the region of the candidate DNMs can be difficult, especially for candidates located in repeat regions. Furthermore, most Sanger resequencing is aimed at validating the heterozygous state of the offspring, not the homozygous state of the parents. If all candidate DNMs are successfully validated, the FPs can be removed from the set of candidate DNMs. However, it is often the case that we cannot check every candidate DNM. In these cases, it is common to estimate the FDR from a subset of candidates that can be checked. The FDR can be estimated as:

$$FDR = \frac{PCR_{failed}}{PCR_{validated} + PCR_{failed}},$$

with $PCR_{validated}$ being the number of candidate DNMs successfully amplified and passing the resequencing validation and $PCR_{failed}$ being the number of candidate DNMs successfully amplified but failing the resequencing validation. We can then adjust the total number (nb) of DNMs in the entire dataset by using the following relationship:

$$nb_{candidateDNMscorrected} = nb_{candidateDNMs} \times (1 - FDR),$$

where $nb_{candidateDNMscorrected}$ is the updated number of DNMs in the dataset. Of note, some studies refer to a FP rate instead of the FDR (e.g., *Bergeron et al., 2021a*; *Jónsson et al., 2017*; *Wang et al., 2020*), yet, it also refers to the ratio of FP calls on the total number of candidates (i.e., true positives and FPs).

A second method to check candidate DNMs is manual curation, using visualization software such as the Integrative Genome Viewer (*Robinson et al., 2011*). By comparing the read mappings of the parents and their offspring at candidate DNMs, FP calls can be detected. Estimates of the FDR using this approach have varied widely depending on the study design, from 91% (*Pfeifer, 2017*) at low coverage to 35% (*Smeds et al., 2016*) at medium coverage to 11% at high coverage (*Bergeron et al., 2021a*). Further work is needed to ensure that manual curation is consistent when applied by different researchers working in different systems.

A third method to estimate the FDR, based on deviations from the expected 50% transmission rate of DNMs to the next generation, can be used if an extended pedigree is available. With this method, *Wu et al., 2020* estimated an FDR of 18%. However, such a deviation from 50% can arise from the expected variance of a binomial distribution, especially if the number of mutations is small. Moreover, clusters of mutations could increase this variance if linked mutations are passed on together to the next generation, especially if the number of trios is small. When this method is used, transmission should be clearly defined as it can be when the grandchild has been genotyped as heterozygote with the mutant allele or alternatively when at least a few reads contain the mutant allele. *Jónsson et al., 2017* used multiple individuals and haplotype sharing to assess the consistent segregation of DNM allele in the next generation.

A fourth method of estimating the FDR takes advantage of monozygotic twins. Germline mutations transmitted from parents to monozygotic twins are expected to be present in both twins as they are derived from the same zygote. *Jónsson et al., 2017* exploited the discordance between candidate DNMs in monozygotic twins to derive the FDR (3%). This estimate is an upper bound because discordance between monozygotic twins is a combination of post-zygotic mutations and FP calls. However, the authors analyzed a unique dataset of 91 human trios with monozygotic twins – data that will be hard to obtain in most species.

### Step 5: Mutation rate estimation

To calculate a per-site per-generation mutation rate, the total number of candidate DNMs (corrected for FPs) should be divided by the number of sites in the genome with full detection power. The denominator takes into account the callable genome (CG) – sites where mutations could have been detected – and the FNR – the rate at which actual DNMs have been missed by the pipeline that has been applied to this point. Assuming that the rate of mutation is similar in the remaining part of the genome, the mutation rate per-site per-generation $\mu$ of a diploid species can be estimated as

$$\mu = \frac{nb_{candidateDNMs} \times (1-FDR)}{2 \times CG \times (1-FNR)}.$$

### Callable genome

Different methods have been used to estimate the CG, the number of sites where a DNM would have been detected if it was there (*Supplementary file 1a*). Many studies used the strict individual filters applied during the detection of candidate DNMs, including all sites where the parents were homozygotes for the reference allele and each individual met the DP, GQ, and any other filters. However, the set of filters and input files used to infer CG differ between studies (*Supplementary file 1a*) and, consequently, estimates vary widely from CG representing 45% (*Tatsumoto et al., 2017*) to 91.5% (*Malinsky et al., 2018*) of the total genome. For instance, some studies used GATK's CallableLoci tool (*Van der Auwera et al., 2013*) that estimates the number of sites that pass the DP filters from the read alignment (.bam) files (e.g., *Wu et al., 2020*) while another study (*Wang et al., 2020*) used the .vcf from the SAMtools mpileup caller (*Li et al., 2009*). From GATK 4 onward, CallableLoci is no longer supported, yet, with the BP_RESOLUTION mode, every single site of the genome has a depth and genotype quality value that can be used to estimate the callable sites (used in e.g., *Bergeron et al., 2021a*; *Pfeifer, 2017*). Moreover, some studies restrict the CG to the orthologous genome in order to match for base composition when making comparisons across species (e.g., *Wu et al., 2020*). Due to these differences, it is important to report which methodology and filters are used to estimate CG.

## Assessing false-negatives

On the number of sites considered callable, additional corrections for the FNR can be included. Indeed, even if the CG represents the sites that pass most of the individual filters, some filters can simply not be applied to non-polymorphic sites. The methods and results differ between studies, with an estimated FNR from 0% (*Smeds et al., 2016*; *Tatsumoto et al., 2017*) to 44% (*Thomas et al., 2018*).

One way to estimate an FNR is to introduce random DNMs to the sequencing reads and run the entire pipeline (steps 2–4) to calculate its efficiency in finding these simulated DNMs (e.g., *Feng et al., 2017*; *Jónsson et al., 2017*; *Pfeifer, 2017*; *Wu et al., 2020*). The FNR can then be estimated as

$$FNR = \frac{denovo_{missed}}{denovo_{simulated}}$$

This method corrects for errors during alignment, post-alignment processing, calling, and filtering as the reads are passed into the pipeline a second time. However, it can be computationally intensive as variant calling needs to be run multiple times and is a resource- and time-intensive step.

Another way to estimate the FNR is to use the number of callable sites that will be filtered away by filters different from those taken into account in the CG estimation, such as site or allelic balance filters (*Bergeron et al., 2021a*; *Besenbacher et al., 2019*; *Thomas et al., 2018*). As some site filters are inferred during variant calling based on statistical tests following known null distributions, it is possible to estimate the proportion of callable sites filtered away by these site filters (*Bergeron et al., 2021a*; *Besenbacher et al., 2015*). Moreover, some true DNMs could have an allelic balance outside the allelic balance filter chosen due to sequencing variability or mosaicism. This bias can be estimated by the heterozygous sites in the offspring (that are not DNMs) presenting an allelic balance outside the allelic balance filter, assuming that this bias occurs at the same rate at DNMs and heterozygous sites in the offspring (i.e., one parent is homozygous for the reference allele, one parent is homozygous for the alternative allele, and the offspring heterozygous). Therefore, FNR can be inferred as the proportion of true heterozygous sites outside the AB filter as

$$FNR = \frac{True\,heterozygous\,sites_{outside\,AB}}{True\,heterozygous\,sites} .$$

Finally, the denominator can be estimated based on a probability to detect a DNM at a site, given various parameters at that site. Thus, there is no clear distinction between *CG* and FNR as the latter is part of the *CG* estimation. Specifically, *Besenbacher et al., 2019* used inherited variants to estimate the probability that a DNM at a given site would pass all filters conditional on the depth of each individual. They then summed these probabilities to calculate the number of callable sites in the genome.

## Mutationathon: Twofold variation in estimated rates from the same trio

To understand the effect of various methods on mutation rate estimates from a single dataset, a three-generation pedigree of rhesus macaque (*M. mulatta*) was analyzed by researchers from five groups: Lucie Bergeron (LB), Søren Besenbacher (SB), Cyril Versoza (CV), Tychele Turner (TT), and Richard Wang (RW). The macaque pedigree consisted of Noot (father), M (mother), Heineken (daughter), and Hoegaarde (Heineken's daughter) (*Figure 3a*). Each individual was sequenced with BGISEQ-500 at an average coverage between 40× (Noot) and 70× (all other individuals). The raw data were trimmed using SOAPnuke (*Chen et al., 2018*) to remove adaptors, low-quality reads, and N-reads (see Materials and methods for more information). Trimmed reads were shared with all participants, who applied their respective pipelines to identify DNMs in Heineken and to estimate a per-site per-generation germline mutation rate.

Each group of investigators implemented their own set of filters (*Table 2—source data 1*) and detected between 18 (CV) and 32 (SB) candidate DNMs. After PCR amplification and Sanger sequencing validation of the DNM candidates from all research groups (43 distinct sites), we validated 33 positions as true-positive DNMs, 6 were determined to be FP calls, and 4 did not successfully amplify (*Figure 3b*, *Figure 3—source data 1*). No group found all true-positive DNMs. Of the 33 true-positive DNMs, only 7 were detected by all research groups (*Figure 3c*). Fourteen additional true-positive mutations were detected by at least four groups; six detected by all except CV, four by all except RW, two by all except LB, one by all except SB, and one by all except TT. Of the 12 remaining true-positive mutations, 5 were detected by three groups, 1 by two groups, and 6 by a single group.

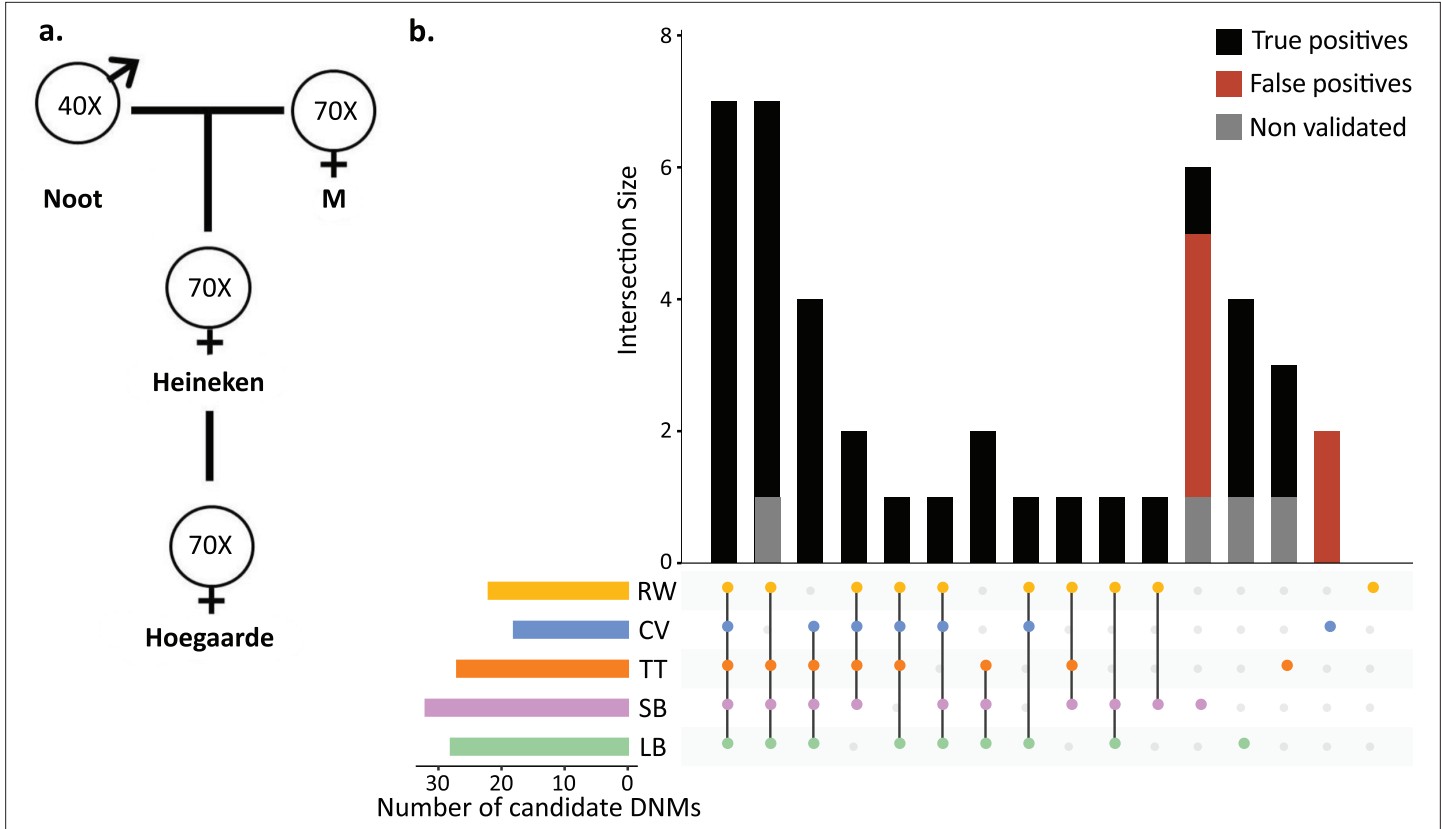

**Figure 3.** Candidate de novo mutations (DNMs) from the Mutationathon. (**a**) The pedigree of three generations of rhesus macaques was sequenced and shared with five groups of researchers. Sequencing coverage is indicated for each individual. (**b**) Upset plot of the 43 candidate DNMs found in Heineken by each research group (LB: Lucie Bergeron; SB: Søren Besenbacher; CV: Cyril Versoza; TT: Tychele Turner; RW: Richard Wang) detected a total of 43 candidate DNMs in Heineken. The first six vertical bars are the candidates shared by at least four different groups. The PCR amplification and Sanger sequencing validation showed that 33 candidates were true-positive DNMs, 6 were false-positive calls (red bars), and 4 did not successfully amplify (gray bars). See Materials and methods for details on the experiment and *Figure 3—source data 2* for the results of the PCR experiment.

The online version of this article includes the following source data and figure supplement(s) for figure 3:

**Source data 1.** PCR validation of the candidate DNMs found by the various pipelines during the Mutationathon.

**Source data 2.** Sanger sequencing chromatograms of the 39 DNM candidate sites that were successfully amplified for the four individuals, i.e. father (Noot), mother (M), offspring (Heineken), and second-generation offspring (Hoegaarde).

**Figure supplement 1.** Mutation spectrum of the trio of rhesus macaques.

The candidate DNMs found by a single group are more likely to be FPs as the six FP candidates revealed by the PCR experiment were all candidates detected by a single pipeline. The differences in candidate DNMs led to differences in the spectrum of mutations (*Figure 3—figure supplement 1*), yet the transition-to-transversion ratio (ti/tv) did not significantly differ between groups (ti/tv all true-positives = 2.7; SB = 2.25; CV = 3; TT = 3.2; RW = 3.2; LB = 5.5; Fisher's exact test p-value=0.87). The transmission rate to the next generation varied between 52% (with SB pipeline: 15 true-positive DNMs transmitted on 29 true-positive candidates) and 67% (with RW pipeline: 14 true-positive DNMs transmitted on 21 true-positive candidates). The transmission rate of all true-positive DNMs (33) was 67% with 21 DNMs transmitted to the next generation; this rate is not significantly different from the expected 50% inheritance (binomial test p-value=0.08).

In addition to identifying DNMs, each group was tasked with estimating the per-site per-generation rate of mutation. The final estimated rate depends on the size of the CG considered by each group, as well as corrections for FPs and false-negatives. Even with the variation in the number of candidate DNMs from each group (*Figure 4a*), different values for these additional parameters could still have resulted in equivalent rate estimates between different groups. However, differences in methodology led to almost a twofold variation in the estimated rates, greater than the variation in the number of

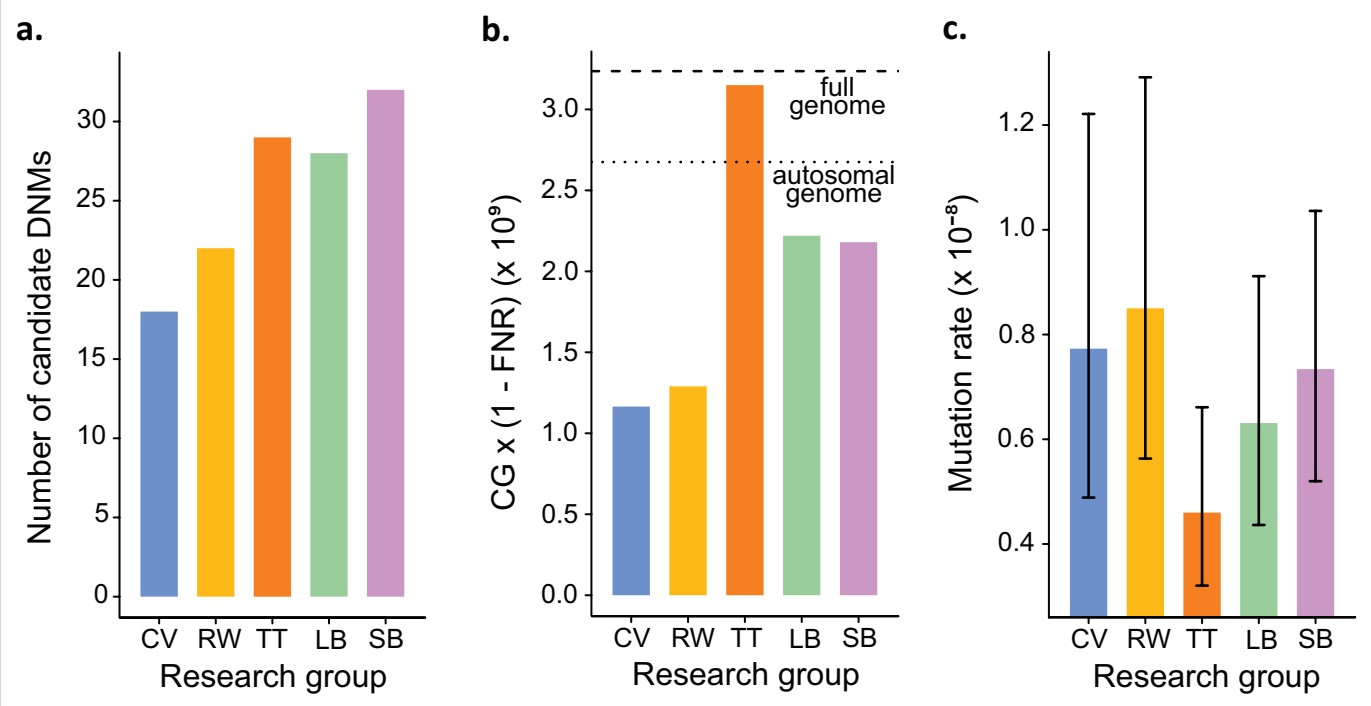

**Figure 4.** Estimated germline mutation rates from the Mutationathon. (a) Number of candidate de novo mutations (DNMs) found by each group (Tychele Turner found two candidates on a sex chromosome). (**b**) Estimation of the denominator (i.e., the callable genome corrected by the false-negative rate [FNR]) by each group. (**c**) Estimated mutation rate per site per generation, the error bars correspond to the confidence intervals for binomial probabilities (calculated using the R package 'binconf').

The online version of this article includes the following source data for figure 4:

**Source data 1.** Number of candidate DNMs, estimated callable genome and per generation mutation rate by each researcher group.

DNMs. TT estimated the lowest rate of $0.46 \times 10^{-8}$ mutations per site per generation (*Figure 4b*). This estimate was based on autosomes and the X chromosome (where two candidates were found), and the CG represented almost the full genome size. Using the full genome size in the denominator is commonly used in human studies, for which most of the genome is callable due to the high-quality reference genome, while stricter corrections are usually applied in nonhuman studies. CV, RW, SB, and LB found similar rates (CV: $0.77 \times 10^{-8}$; RW: $0.85 \times 10^{-8}$; SB: $0.73 \times 10^{-8}$; LB: $0.63 \times 10^{-8}$), with 25% differences between the lowest and the highest rate and large overlap of the confidence intervals (*Figure 4c*). RW estimated the highest rate with $0.85 \times 10^{-8}$ mutations per site per generation, from a relatively small set of candidates (22), yet the denominator was also small as CG represented about 50% of the autosomal genome. SB and LB estimated a similar value of CG, representing approximately 80% of the autosomal genome; however, there was a difference in rates due to the smaller number of candidates found by LB (28) compared to SB (32).

The different individual filters applied by each group explain some of the differences in the candidate DNMs (*Table 2*, *Table 2—source data 1*). For instance, many groups filtered away candidate sites where the parents were heterozygotes as they could be more prone to FP calls. TT's pipeline was the only one to find a candidate mutation at a site where the father was heterozygous C/G, the mother was homozygous for the reference allele G/G, and the offspring was heterozygous A/G. These genotypes were validated by the PCR experiment, indicating that a true germline mutation has arisen at a heterozygous site in a parental genome. Each method varied in power to detect the true DNMs (sensitivity), and in the proportion of validated true candidates on the overall candidates found (precision). For instance, RW used especially conservative filters on the allelic balance for both the offspring (AB) and the number of alternative alleles allowed in the parents (AD). It resulted in a lower sensitivity, only 22 candidates were found, but a high precision as no candidates were determined to be FP calls. Similarly to RW, some groups were conservative on the AB filter, while other groups were more conservative on the GQ filter (SB and LB) or DP filter (LB, CV, RW). For instance, SB used a relaxed filter

**Table 2.** Site-specific and sample-specific filters used by the different groups to detect de novo mutations (DNMs) in Heineken (difference in the other steps of the pipeline in **Table 2—source data 1**).

| Research group | Candidate DNMs | Site-specific filters | Sample-specific filters | Additional filters |
|---|---|---|---|---|
| CV | 18 | GATK Best Practices hard filter criteria | 0.5 × dpind < DP < 2 × dpind<br>GQ > 40<br>AD > 0<br>0.25 < AB < 0.75 | |
| RW | 22 | QD < 2.0<br>MQ < 40.0<br>FS > 60.0<br>SOR > 3.0<br>MQRankSum < −12.5<br>ReadPosRankSum < −8.0 | 20 < DP < 80<br>GQ > 20<br>AD > 0<br>0.35 < AB | Alternative allele on both strands |
| TT | 27 | Remove variants in recent repeats or in homopolymers of AAAAAAAAAA or TTTTTTTTTT | DP > 10<br>GQ > 20<br>AD > 0<br>0.25 < AB | Overlap three different variant callers<br><br>Filter on LCR |
| LB | 28 | QD < 2.0<br>FS > 20.0<br>MQ < 40.0<br>MQRankSum < −2.0<br>MQRankSum > 4.0<br>ReadPosRankSum < −3.0<br>ReadPosRankSum > 3.0<br>SOR > 3.0 | 0.5 × dpind < DP < 2 × dpind<br>GQ > 60<br>AD none<br>0.3 < AB < 0.7 | Manual curation (six candidates removed) |
| SB | 32 | FS > 30.0<br>MQRankSum < −10<br>MQRankSum > 10<br>ReadPosRankSum < −2.5<br>ReadPosRankSum > 2.5<br>BaseQRankSum < −13<br>BaseQRankSum > 13 | 10 < DP < 2× dpind<br>GQ > 55<br>AD > 0<br>0.3 < AB | Alternative allele in both strands. lowQ AD2 > 1 |

LB: Lucie Bergeron; SB: Søren Besenbacher; CV: Cyril Versoza; TT: Tychele Turner; RW: Richard Wang.

The online version of this article includes the following source data for table 2:

**Source data 1.** Details on the methodology and filtering criteria applied by the five different pipelines to estimate the mutation rate on the common pedigree.

on DP, with a minimum threshold of 10×, but a relatively conservative threshold on AB and GQ criteria. TT did not use strict filters for any parameter; however, the precision was increased by the required overlap among multiple variant callers.

We explored the effect of the individual filter on the number of candidate DNMs, the number of FP calls, the CG, the FNR, and the final estimated mutation rate per-site per generation (μ). We used the LB pipeline (see individual filters in **Table 2** and other methods in **Table 2—source data 1**) and changed one filter at a time using various criteria used by the Mutationathon participants and in the literature (**Figure 5**, **Figure 5—source data 1**). The GQ filter had the largest impact on the number of mutations and the final estimated mutation rate. The number of candidate DNMs found with GQ < 20 was three times higher than the one obtained with the most conservative GQ filter ($GQ_{Hom} < 100$ and $GQ_{Het} < 200$), and the difference was still twofold after correcting for FP calls. The CG also decreased with GQ < 80, leading to an estimated rate 39% lower when GQ < 80 ($μ = 0.56 × 10^{-8}$) compared to when GQ < 20 ($μ = 0.91 × 10^{-8}$). This filter also seems to be the most efficient at reducing the number of FP calls, estimated here with the manual curation method, as more than 90% of the candidates DNMs were FPs with no GQ filter while we found no FPs with conservative GQ filters (GQ < 80 and $GQ_{Hom} < 100$ and $GQ_{Het} < 200$). Another important filter was the allelic balance on the heterozygous offspring, resulting in a twofold difference in the number of candidate DNMs detected, and 1.5-fold difference after the correction for FP calls. Yet, the estimated FNR was almost

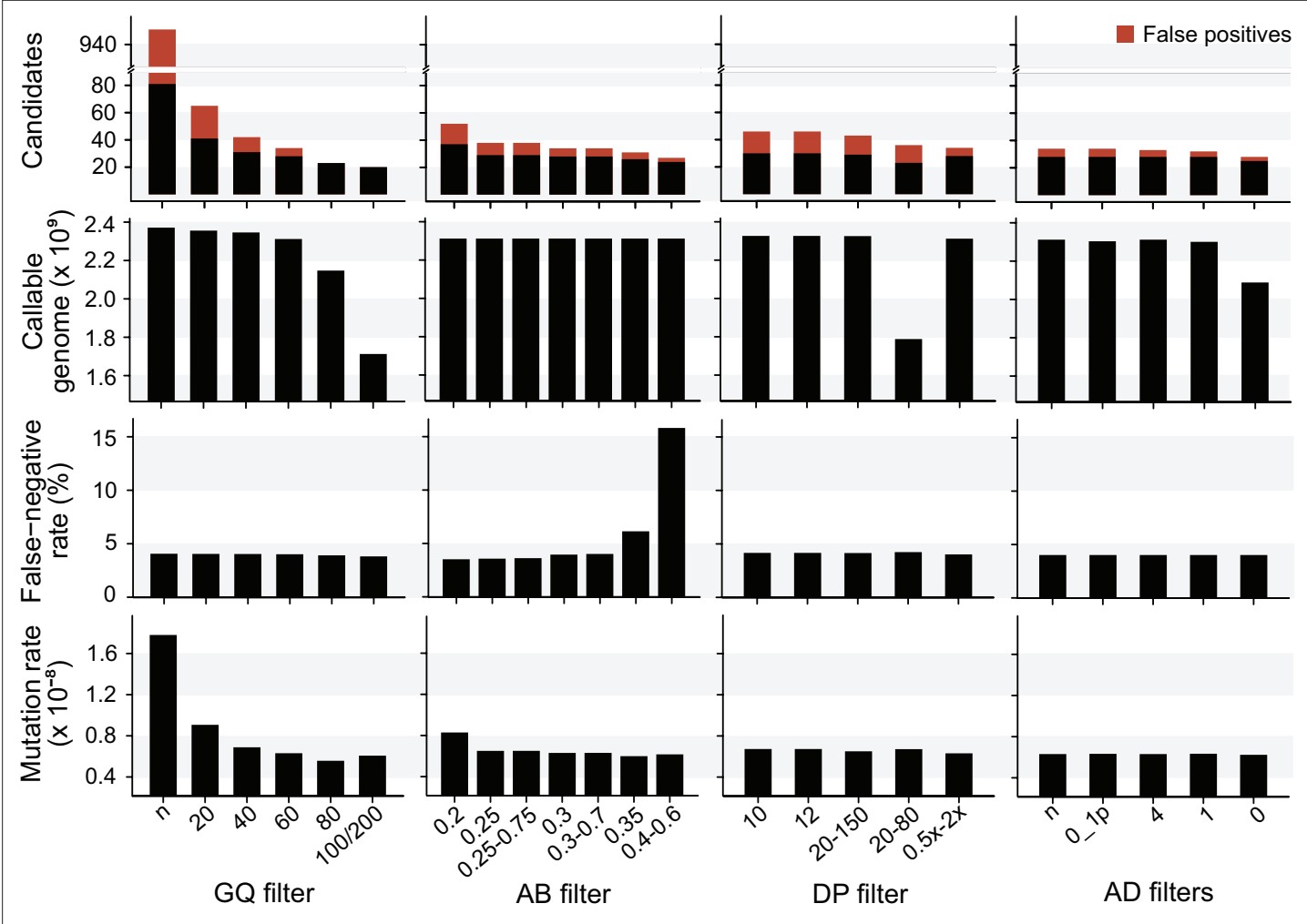

**Figure 5.** The impact of individual filters on the estimated rate of a trio of rhesus macaques. The default filters used by Lucie Bergeron (LB) pipeline were DP < 0.5 × depth$_{individual}$; DP > 2 × depth$_{individual}$; GQ < 60; AB < 0.3; AB > 0.7, no AD filter.

The online version of this article includes the following source data for figure 5:

**Source data 1.** Details on the number of candidate DNMs, the number of false positive calls, the size of the callable genome, the false negative rate and the final estimated mutation rate using various individual filters.

five times higher when using a conservative AB filter (AB < 0.4 and AB > 0.6; FNR = 15.8%) compared to the least conservative AB filter (AB < 0.2; FNR = 3.5%). This led to a mutation rate estimate 28% lower with the conservative AB filter (AB < 0.4 and AB > 0.6; μ = 0.69 × 10$^{-8}$ and AB < 0.2; μ = 0.83 × 10$^{-8}$). The DP filter also impacted the estimated rate but to a lesser extent with only a 6% difference between the estimated mutation rate with DP < 10 (μ = 0.67 × 10$^{-8}$) and the most conservative DP filter (DP <0.5 × depth$_{individual}$ and DP >2 × depth$_{individual}$; μ = 0.63 × 10$^{-8}$). Finally, the AD filter did not show a large impact on the mutation rate, with less than 2% difference between no filter on AD (μ = 0.63 × 10$^{-8}$) and the conservative AD >0 (μ = 0.62 × 10$^{-8}$).

The mutation rate was calculated with LB pipeline as $\frac{nb_{candidateDNMs} - FP}{2 \times CG \times (1 - FNR)}$ .

These results show that some of the differences in estimated rates between the five research groups may be attributed to the individual filters. Yet, earlier steps in the different bioinformatic pipelines could also lead to differences in candidate DNMs and estimated rates. For instance, the site filters were different between some of the groups (see *Table 2—source data 1*). Testing different combinations of site filters on the shared trio of rhesus macaques affected the set of SNPs detected, which could lead to variation in candidate DNMs detected. For instance, on the 12,634,956 variants found by LB pipeline, 473,142 SNPs were removed when using GATK-advised filters (QD < 2.0; MQ <

40.0; FS > 60.0; SOR > 3.0; MQRankSum < –12.5; ReadPosRankSum < –8.0), while the stringent filters used by LB pipeline (QD < 2.0, FS > 20.0, MQ < 40.0, MQRankSum < –2.0, MQRankSum > 4.0, ReadPosRankSum < –3.0, ReadPosRankSum > 3.0, and SOR > 3.0) removed 1,124,005 SNPs. Despite this difference in the number of SNPs, using the LB pipeline to detect candidate DNMs on the three callset (no filters, GATK-advised filter, or stringent filters), led to the same final number of candidate DNMs due to the stringent individual filters applied in the following steps of the pipeline. Other steps, such as mapping and variant calling, could also lead to some of the differences between the five groups. For instance, the six candidates identified as FPs by the Sanger sequencing were filtered away in the LB pipeline. Four of the FP candidates were not detected because all individuals were genotyped as homozygous for the reference allele, one position was filtered out by the mapping quality site filter (MQ < 40), and one position had DP = 0. Thus, differences in the mapping of the reads and variant callers explain some of the discrepancies between pipelines.

Overall, these results show that for the same dataset differences in estimated mutation rates caused by methodological discrepancies are non-negligible. Therefore, such differences should be considered when comparing mutation rates between different species when they are estimated by different pipelines. Some of the differences in estimated rates between groups can be attributed to the different individual filters applied for the detection of candidate DNMs. Most notably, varying the GQ and AB filters leads to large variations in estimated rates. Some of the difference is also introduced in earlier steps when mapping reads and calling variants. Moreover, the estimated CG is different between the five groups; in addition to changing the denominator of the mutation rate calculation, this difference could reflect the ability of individual methods to query mutation in different genomic regions. Some variation might therefore be explained by true mutation rate heterogeneity between genomic regions (such as low- or high-complex regions). Our results also suggest that despite the different methods and filters the estimated rates are comparable when both the numerator (number of candidates and FPs) and the denominator (CG and FNR) are carefully corrected. For instance, CV, SB, LB, and RW estimated similar rates, but SB and RW used a probabilistic method to calculate the CG, while LB used strict filters (DP and GQ) on a base-pair resolution .vcf and corrected for FNR using the site filters and the allelic balance filter and CV used a similar method to estimate CG, yet, did not apply a correction for FNR. Finally, the Mutationathon was carried out using a single trio, thus providing no information about sampling error. It would be of interest to pursue a similar effort of comparing methodologies using larger or more pedigrees in order to reach a broader consensus. Using available human datasets could further allow some control on known benchmarks such as ti/tv or known population SNPs, but may not represent the scenario confronting researchers studying new species.

## Best practices

When estimating germline mutation rates from pedigree samples, there is no standardized set of methods. Different studies use different software versions and filtering thresholds, which can impact the estimated rate and can complicate the comparison of rates between or within species across studies (in addition to the biological variation introduced by the age of the parents used in each study; *Table 1*). Here, we provide guidelines for each step in DNM calling and rate estimation. However, we note that sample quality, reference genomes, and other technical factors differ across studies and thus require study- or species-specific thresholds. Therefore, it is advised to report the methodology used in a standardized way. It would also be helpful to release the .vcf files, which would allow reproducibility and further comparisons on larger datasets. *Table 3* proposes a checklist of parameters that should be reported in studies of germline mutation rates.

Moreover, some benchmarks could be helpful to ease the comparison between studies such as:

- the transition-to-transversion ratio (ti/tv),
- the spectrum of mutations (see *Figure 3—figure supplement 1* for an example from the Mutationathon),
- the percentage of mutations in CpG locations,
- the base composition (percentage of A/T or C/G),
- the nucleotide heterozygosity in unrelated individuals,
- if population data are available, the number of DNMs that are in known SNPs of the population,
- the contribution of each sex to the total number of mutation bias when phasing of mutations is possible,
- the transmission rate to the next generation when extended trios are available,

**Table 3.** Information that should ideally be reported when presenting results on de novo mutations (DNMs).
See *Table 2—source data 1* for an example of this table filled out for the five pipelines used to analyze the trio of rhesus macaques.

| Step of the analysis | Information to report |
| --- | --- |
| 1. Sampling and sequencing | Type of sample (tissue, etc.) |
| | Storage duration, buffer, temperature |
| | Type of library preparation |
| | Average sequencing coverage |
| | Sequencing technology and read lengths |
| 2. Alignment and post-alignment processing | Trimming of adaptors and low-quality reads |
| | Reference assembly version<br>Autosomes only or whole genome? |
| | Mapping software and version |
| | Duplicate removal software and version |
| | Base quality score recalibration (yes/no) |
| | If yes, which type of data used as known variants? |
| | Realignments around indels? |
| | Other filters? |
| 3. Variant calling | Software and version |
| | Mode: joint genotyping? GVCF blocks? GVCF in base-pair resolution? |
| 4. Detecting DNMs | Site filters on .vcf files and justification |
| | Individual filters, threshold, and remaining candidates after each filter |
| | False discovery rate estimation method: PCR validation? Manual curation? Transmission rate deviation? Removal of low-complexity regions, cluster mutations, or recurrent mutations? |
| 5. Mutation rate estimation | Callable genome estimation method: File used? Filters taken into account? |
| | False-negative rate estimation method: simulation? Filters? Probability? |

- the average age of the parents at the time of reproduction, if known
- the distribution of the allelic balance of true heterozygotes, candidate DNMs after all filters except the allelic balance, and the final set of candidate DNMs.

## Conclusion and perspectives

Different filters can lead to differences in estimated rates, which emphasizes the difficulty in comparing pedigree-based germline mutation rates estimated from different studies. The variation observed could be partially due to the biology and life-history traits of species, but some of the variations will also be caused by methodological differences. Here, we provided some best practices that can be used when estimating germline mutation rates from pedigree samples. However, it is hard to provide hard cutoffs of filters that apply to every situation, and we advise choosing appropriate filters depending on the data available. We have also raised some points that should be addressed in individual studies, such as estimation of the FDR, FNR, and the CG size. Nevertheless, more exploration should be done to understand the best strategy for the different steps required in every study of the mutation rates. Without a clear consensus on approaches for estimating the germline mutation rate, it seems that the best strategy will be to carefully report all methods and parameters used. The trio of

rhesus macaque used in this analysis is publicly available, along with the validated candidate DNMs, and could serve as a resource for testing new strategies. On a more positive note, it is important to point out that two recent, independent studies of the per-generation mutation rate in rhesus macaque reported rates that were within 5% of each other for individuals of the same age (*Bergeron et al., 2021a*; *Wang et al., 2020*). We hope that careful studies using a variety of methods will be able to similarly arrive at accurate estimates of important biological parameters.

With the growing number of studies on pedigree-based estimation of germline mutation rate, some directions that have been neglected could be explored. For instance, even when the sample size is large, most studies use samples originating from small geographic regions; it would be of great interest to further explore potential variation in mutation rates across diverse populations (e.g., *Kessler et al., 2020*). A large study with many trios sequenced using the same protocol also provides more information about the features that separate true and false variant calls. If the number of sequenced samples is sufficiently large, it might even become feasible to estimate a site-specific error rate for each position in the genome. Such improved error rate estimates would further improve the ability to avoid FP DNM calls.

Most studies are conducted on genomic DNA collected from somatic tissues. As a result, if samples come from only a single trio, one cannot distinguish early postzygotic mutations occurring in the offspring from germline mutations in the parents. While mutations occurring early enough in offspring development will be passed on to the next generation – and should therefore still be considered DNMs – they will behave differently from mutations arising in the parental generation. For instance, we will not expect an increase of these mutations with parental age (*Jónsson et al., 2018*). Therefore, it is of interest to distinguish between these two types of mutation, especially for biomedical research. A possible way to discard those mutations would be to compare somatic and germline cells from the same individual. However, extracting DNA directly from sperm and eggs can be challenging, especially for nonhuman species, limiting the application of this strategy. Another area for additional future work is to look at de novo structural variants. As they are even rarer than SNPs, it is hard to detect them over a single generation. Yet, with the growing number of trios and generations considered in recent studies, it would be of interest to quantify and describe those DNMs as well (e.g., *Belyeu et al., 2021*; *Thomas et al., 2021*). The development of accurate long-read sequencing technologies also offers opportunities for better detection of DNMs and de novo structural variants. Finally, most studies on nonhuman species only explore the autosomal chromosomes, largely because important filters such as allelic balance cannot be used on the sex chromosomes in both sexes. However, given the consistent differences observed between species in the rate of evolution on autosomes and sex chromosomes (e.g., *Wilson Sayres and Makova, 2011*), it would be very interesting to look more closely at the per-generation mutation rate on sex chromosomes.

## Materials and methods
### Mutationathon sequences

The pedigree used for the Mutationathon was previously sequenced as part of a larger project on the mutation rate of rhesus macaques (BioProject: PRJNA588178; *Bergeron et al., 2021a*). Nine lanes were used in this analysis (three lanes for the father and two lanes for the other individuals) and are publicly available on NCBI:

1. CL100066413_L01 (SRA run SRR10426295), mother M
2. CL100089164_L01 (SRA run SRR10426294), mother M
3. CL100078308_L01 (SRA run SRR10426275), father Noot
4. CL100078335_L01 (SRA run SRR10426264), father Noot
5. CL100078335_L02 (SRA run SRR10426253), father Noot
6. CL100066412_L02 (SRA run SRR10426291), offspring Heineken
7. CL100095002_L02 (SRA run SRR10426290), offspring Heineken
8. CL100066408_L01 (SRA run SRR10426256), next-generation offspring Hoegaarde
9. CL100094917_L01 (SRA run SRR10426255), next-generation offspring Hoegaarde

A trimming step was done on all sequences to remove the adaptors (allowing a mismatch of two bases), the low-quality reads (with more than 5% of N bases or a base quality score < 10 in more than 20% of the read), and the reads smaller than 60 bases after the quality control. Trimming was done

using SOAPnuke (RRID:SCR_015025) version 1.5.6 (*Chen et al., 2018*), with the following command: *>SOAPnuke filter -f AAGTCGGAGGCCAAGCGGTCTTAGGAAGACAA -r AAGTCGGATCGTAGCC ATGTCGTTCTGTGAGCCAAGGAGTTG –1 sequence_read_1–2 sequence_read_2 -Q 2l 10 -q 0.2 -E 60–5 0M 2 -o sequence_clean -C sequence_read_1_clean -D sequence_read_2_clean.*

Each group implemented its pipeline to estimate a rate (details are provided in *Table 2—source data 1*).

## Data analysis

The comparison of each individual filter was done using LB pipeline, changing one filter at a time and recalculating the number of candidates DNMs detected, the potential FP candidates with the manual curation method, the CG, the FNR on the allelic balance filter and site filters, and the mutation rate per site per generation. The comparison of the site filters was also done on the SNPs found by LB pipeline.

## PCR experiment and Sanger resequencing

We designed multiple sets of primers for the 43 candidate sites on NCBI primer blast tool (*Ye et al., 2012*: https://www.ncbi.nlm.nih.gov/tools/primer-blast/, RRID:SCR_003095). In some cases, sequencing primers were adjusted to avoid sequencing failure due to poly-AAA or TTT runs. PCRs were carried out in 25 µL volumes (2.5 units Dream Taq DNA Polymerase [Thermo Scientific], 1X Dream Taq Green Buffer, 0.2 mM dNTPs, 2–3 mM MgCl$_2$, 2.5–44 ng DNA template, filled to 25 µL with double-distilled [ddH$_2$O] water). Thermocycling was performed in a Bio-Rad PTC-100 thermocycler. The cycle program comprised an initial denaturation at 95°C for 2 min, followed by 35 cycles of 15 s at 95°C, 15 s at 52–55°C, and 30 s at 72°C. Cycling was terminated with a 5 min extension at 72°C. PCR products were purified using commercially available spin columns (Invitek) or PureIT ExoZap PCR Clean-up (Ampliqon). Sanger sequencing was conducted at Eurofins Genomics, Europe, using the primers of the amplification procedure using both forward and reverse primers. In *Figure 3—source data 2*, the chromatograms with the best base quality value are provided. *Supplementary file 1d* provides details about the primers and accession number of the sequences on GenBank.

## Data and code availability

All the sequences used for the Mutationathon were previously generated and released in NCBI (*Bergeron et al., 2021a*). The sequences used were for the mother M (BioSample SAMN13230631): lanes CL100066413_L01 (SRA run SRR10426295) and CL100089164_L01 (SRA run SRR10426294); for the father Noot (BioSample SAMN13230623): lanes CL100078308_L01 (SRA run SRR10426275), CL100078335_L01 (SRA run SRR10426264), and CL100078335_L02 (SRA run SRR10426253); for the offspring Heineken (BioSample SAMN13230633): lanes CL100066412_L02 (SRA run SRR10426291) and CL100095002_L02 (SRA run SRR10426290); and for the second-generation offspring Hoegaarde (BioSample SAMN13230649): lanes CL100066408_L01 (SRA run SRR10426256) and CL100094917_L01 (SRA run SRR10426255). The Sanger sequences generated during the PCR validation were deposited on GenBank under the accession numbers MZ661796–MZ662076.

The scripts used by the participants of the Mutationathon are publicly available:

- CV: https://github.com/PfeiferLab/mutationathon (*Versoza, 2021*)
- RW: https://github.com/Wang-RJ/mutationathon (*Wang, 2021*)
- TT: *Wilfert et al., 2021*
- LB: https://github.com/lucieabergeron/germline_mutation_rate (*Bergeron, 2021b*)
- SB: https://github.com/besenbacher/GreatApeMutationRate2018 (*Besenbacher, 2019*)

## Acknowledgements

We would like to thank GenomeDK at Aarhus University and Arizona State University's Research Computing for providing computational resources and support for the LB pipeline and CV pipeline, respectively. We also thank Hákon Jónsson, the editor George Perry, the reviewer Aaron R Quinlan, as well as two additional anonymous reviewers for helpful comments on the manuscript and Maria Kamilari for helpful input on the PCR validation experiment. SPP was supported by a US National Science

Foundation CAREER grant (DEB-2045343). LB was supported by a Carlsberg Foundation Grant to GZ (CF16-0663).

## Additional information

### Funding

| Funder | Grant reference number | Author |
|---|---|---|
| Carlsbergfondet | CF16-0663 | Guojie Zhang |
| National Science Foundation | CAREER DEB-2045343 | Susanne P Pfeifer |
| Marie Skłodowska-Curie Actions | 840414, MutANTs | Hwei-yen Chen |

The funders had no role in study design, data collection and interpretation, or the decision to submit the work for publication.

### Author contributions

Lucie A Bergeron, Conceptualization, Formal analysis, Methodology, Visualization, Writing - original draft, Writing – review and editing; Søren Besenbacher, Conceptualization, Formal analysis, Methodology, Writing - original draft, Writing – review and editing; Tychele Turner, Cyril J Versoza, Richard J Wang, Formal analysis, Methodology, Writing – review and editing; Alivia Lee Price, Lab experiment, Methodology, Writing – review and editing; Ellie Armstrong, Meritxell Riera, Jedidiah Carlson, Hwei-yen Chen, Matthew W Hahn, Kelley Harris, April Snøfrid Kleppe, Elora H López-Nandam, Priya Moorjani, Susanne P Pfeifer, George P Tiley, Anne D Yoder, Methodology, Writing – review and editing; Guojie Zhang, Mikkel H Schierup, Conceptualization, Methodology, Supervision, Writing - original draft, Writing – review and editing

### Author ORCIDs

Lucie A Bergeron (ID) http://orcid.org/0000-0003-1877-1690
Søren Besenbacher (ID) http://orcid.org/0000-0003-1455-1738
Ellie Armstrong (ID) http://orcid.org/0000-0001-7107-6318
Matthew W Hahn (ID) http://orcid.org/0000-0002-5731-8808
Kelley Harris (ID) http://orcid.org/0000-0003-0302-2523
April Snøfrid Kleppe (ID) http://orcid.org/0000-0001-7866-3056
Susanne P Pfeifer (ID) http://orcid.org/0000-0003-1378-2913
George P Tiley (ID) http://orcid.org/0000-0003-0053-0207
Mikkel H Schierup (ID) http://orcid.org/0000-0002-5028-1790

### Decision letter and Author response

Decision letter https://doi.org/10.7554/eLife.73577.sa1
Author response https://doi.org/10.7554/eLife.73577.sa2

## Additional files

### Supplementary files

• Supplementary file 1. Four supplementary tables with details on the methods used in the literature, Genome Analysis ToolKit (GATK) site filters, site-specific and sample-specific filters used in the literature, and the PCR experiment.

• Transparent reporting form

### Data availability

The sequences of the pedigree analyzed are available on NCBI under the accession numbers: SRR10426295, SRR10426294, SRR10426275, SRR10426264, SRR10426253, SRR10426291, SRR10426290, SRR10426256, SRR10426255. The PCR experiment and Sanger resequencing produced for this work are deposited on GenBank under the accession number MZ661796–MZ662076.

Supplementary file 1 describe the data. The scripts used by the participants of the Mutationathon are publically available on different github described in the manuscript. Figure 3, 4 and 5 can be reproduced with the data in Figure 3—source data 1, Figure 4—source data 1, and Figure 5—source data 1 .

The following datasets were generated:

| Author(s) | Year | Dataset title | Dataset URL | Database and Identifier |
|---|---|---|---|---|
| Bergeron LA | 2021 | The mutationathon highlights the importance of reaching standardization in estimates of pedigree-based germline mutation rates | https://www.ncbi.nlm.nih.gov/sra/SRR10426295 | NCBI Sequence Read Archive, SRR10426295 |
| Bergeron LA | 2021 | The mutationathon highlights the importance of reaching standardization in estimates of pedigree-based germline mutation rates | https://www.ncbi.nlm.nih.gov/sra/SRR10426294 | NCBI Sequence Read Archive, SRR10426294 |
| Bergeron LA | 2021 | The mutationathon highlights the importance of reaching standardization in estimates of pedigree-based germline mutation rates | https://www.ncbi.nlm.nih.gov/sra/SRR10426275 | NCBI Sequence Read Archive, SRR10426275 |
| Bergeron LA | 2021 | The mutationathon highlights the importance of reaching standardization in estimates of pedigree-based germline mutation rates | https://www.ncbi.nlm.nih.gov/sra/SRR10426264 | NCBI Sequence Read Archive, SRR10426264 |
| Bergeron LA | 2021 | The mutationathon highlights the importance of reaching standardization in estimates of pedigree-based germline mutation rates | https://www.ncbi.nlm.nih.gov/sra/SRR10426253 | NCBI Sequence Read Archive, SRR10426253 |
| Bergeron LA | 2021 | The mutationathon highlights the importance of reaching standardization in estimates of pedigree-based germline mutation rates | https://www.ncbi.nlm.nih.gov/sra/SRR10426291 | NCBI Sequence Read Archive, SRR10426291 |
| Bergeron LA | 2021 | The mutationathon highlights the importance of reaching standardization in estimates of pedigree-based germline mutation rates | https://www.ncbi.nlm.nih.gov/sra/SRR10426290 | NCBI Sequence Read Archive, SRR10426290 |
| Bergeron LA | 2021 | The mutationathon highlights the importance of reaching standardization in estimates of pedigree-based germline mutation rates | https://www.ncbi.nlm.nih.gov/sra/SRR10426256 | NCBI Sequence Read Archive, SRR10426256 |
| Bergeron LA | 2021 | The mutationathon highlights the importance of reaching standardization in estimates of pedigree-based germline mutation rates | https://www.ncbi.nlm.nih.gov/sra/SRR10426255 | NCBI Sequence Read Archive, SRR10426255 |

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
