## [Editor Report]

Bergeron et al. show that mutation rate independently estimated by several teams with the same pedigree dataset can be different due to the methods and approaches used to identify de novo mutations. This result is of primary importance because it shows the necessity to have standard mutation identification methods and the difficulties to compare mutation rates from different studies.

---

## [Decision Letter]

**Decision letter after peer review:**

Thank you for submitting your article "Mutationathon: towards standardization in estimates of pedigree-based germline mutation rates" for consideration by *eLife*. We are sorry for the long review process. Given the long list of authors, it was difficult for us to find reviewers with no conflict of interest. Your article has now been reviewed by 3 peer reviewers, and the evaluation has been overseen by a Reviewing Editor and George Perry as the Senior Editor. The following individual involved in review of your submission has agreed to reveal their identity: Aaron R. Quinlan (Reviewer #3).

Essential revisions:

The authors should address issues or add discussion on de novo mutations calling in multi-sibling families, based on multi-generational pedigrees or on multiple tissues, the cellular processes causing germline de novo mutations which can vary between species, and the effect of the tissue on somatic mutation contamination, that could require adjustments in the method. Furthermore, the authors should discuss the fact that there is no sampling error at play in the mutationathon, since all teams used the same input data.

Since a gold standard approach is not achievable, the authors may want to revise the title.

Finally, the text should be made more accessible to a broad audience.

*Reviewer #1 (Recommendations for the authors):*

Page 16 line 428-429:

According to the authors, the multiple nucleotide mutations (MNM) are removed from most of studies because believed to be false. But some elements also seem to show that this type of MNM are possible, in example when replication-transcription machineries overlap. Do the authors have any example of MNM tested by PCR in any cited study? This could help to choose between keep or discard. If necessary, see for example CurrBiol. 2011 Jun 21;21(12):1051-4. doi: 10.1016/j.cub.2011.05.013.

Page 23 lines 611-630.

The exact value of µ is provided only for two teams (TT and RW). Can the authors also add the exact values of µ from other teams somewhere?

Page 25:

The table 2 looks uncomplete to me, is there MQ different threshold between the teams that need to be mentioned?

Page 30 line 754:

1) The authors should add a table (even in supplementary) with all mutation candidates, indicating

Chromosome position ref alt team true/false (and other relevant information at their discretion). Maybe the low complexity region led to higher rate of mutation identification candidates, where the 5 teams have so many different mutation candidates, with only 7 true shared. Complexity around mutation candidate positions may be addressed by several methods (see https://doi.org/10.1093/molbev/msab140). This point can be to the benchmarks at the end of page 30 if relevant.

2) For me, an important implication of this study is the mutation spectrum. The 5 mutations rates estimates are of the same order (one is still high), but the mutation candidates between the teams differ. This means that an analysis of the mutation spectrum and intragenomic mutation rate variation between the teams can be different, more significantly than the difference in mutation rate itself: AT bias, insertion/deletion, chromosome mutation rate, and so one. I highly encourage the authors to realize a later study exploring in detail the benchmarks proposed at the end of page 30.

*Reviewer #2 (Recommendations for the authors):*

I had couple of suggestions:

It would be very useful to introduce a site-specific error rate parameter at least for the human data.

Also it would be interesting if the authors can use some of their suggested parameters: alignment, variant calling and post-filtering on some of the available data to show how their guidelines may improve the mutation rate estimate.

*Reviewer #3 (Recommendations for the authors):*

Assessing the variability of germline mutation rates predicted from each pipeline via application to more (e.g.) pedigrees would be informative.

LCRs on line 434 refer to low-complexity regions identified by Heng Li as drivers of false-positive variant predictions.

The color scheme in Figure 3B will be difficult for the color blind.

Figure 3C is difficult to parse. Perhaps consider an "Upset" plot.

Do the GQ filters discussed on lines 670-674 apply to each member of the pedigree? That is, is GQ>=20 applied to the genotype for each macaque at each site?

Figure 5 is very difficult to read. Please make the axis labels larger. You may be able to also make the figure more legible by packing the bars more tightly so as to waste less whitespace.

The manuscript is often difficult to read with all of the VCF acronyms used to describe the filtering logic applied. While decipherable for those familiar with the "joys" of VCF, you might consider a more accessible approach for a broader audience.

---

## [Author Response]

The authors should address issues or add discussion on de novo mutations calling in multi-sibling families, based on multi-generational pedigrees or on multiple tissues, the cellular processes causing germline de novo mutations which can vary between species, and the effect of the tissue on somatic mutation contamination, that could require adjustments in the method. Furthermore, the authors should discuss the fact that there is no sampling error at play in the mutationathon, since all teams used the same input data.

We have now extended the discussion on the following parts: multi-sibling families (in the section on sample size), multi-generation pedigrees (section on sample size), and multiple tissues (section on sample type). We further extended the section on how to detect somatic mutations in cases of tissue contamination (section on sample type). We also now mention the limit of the Mutationathon with regard to sampling error, and have added a brief discussion of the advantages to studying larger/more pedigrees in a similar comparison in the future. We agree that there are likely to be some cellular processes that differ between species, however we do not believe that this would change the conclusions of our study.

Since a gold standard approach is not achievable, the authors may want to revise the title.

We have now revised the title for “Mutationathon: the importance of reaching standardization in estimates of pedigree-based germline mutation rates”

Finally, the text should be made more accessible to a broad audience.

We agree that especially the filtering part was a bit hard to read for people not used to the VCF format. Therefore we used a box to explain the jargon and make it more accessible.

Reviewer #1 (Recommendations for the authors):Page 16 line 428-429:According to the authors, the multiple nucleotide mutations (MNM) are removed from most of studies because believed to be false. But some elements also seem to show that this type of MNM are possible, in example when replication-transcription machineries overlap. Do the authors have any example of MNM tested by PCR in any cited study? This could help to choose between keep or discard. If necessary, see for example CurrBiol. 2011 Jun 21;21(12):1051-4. doi: 10.1016/j.cub.2011.05.013.

We agree with the reviewer that this assumption can be erroneous as there is some evidence now of MNMs. We have now extended this part adding citation in which MNMs have been tested with PCR (page 18 line 458-464).

Page 23 lines 611-630.The exact value of µ is provided only for two teams (TT and RW). Can the authors also add the exact values of µ from other teams somewhere?

We have now added this information for each team in the main text (page 25 lines 645).

Page 25:The table 2 looks uncomplete to me, is there MQ different threshold between the teams that need to be mentioned?

A more complete table of all differences in methods is presented in Table 2 – source data. But we agree that MQ and other site-specific filters may be worth mentioning in the Table 2. We have now modified it accordingly (pages 27 and 28)

Page 30 line 754:1) The authors should add a table (even in supplementary) with all mutation candidates, indicatingChromosome position ref alt team true/false (and other relevant information at their discretion). Maybe the low complexity region led to higher rate of mutation identification candidates, where the 5 teams have so many different mutation candidates, with only 7 true shared. Complexity around mutation candidate positions may be addressed by several methods (see https://doi.org/10.1093/molbev/msab140). This point can be to the benchmarks at the end of page 30 if relevant.

We agree with the reviewer on the necessity of such a table and have already uploaded it as source_data_fig3. Indeed, TT had a filter on LCR, this has now be added on Table 2 (pages 27 and 28). Moreover, we agree that this filter should be in the benchmark and have now added this in Table 3 (page 33).

2) For me, an important implication of this study is the mutation spectrum. The 5 mutations rates estimates are of the same order (one is still high), but the mutation candidates between the teams differ. This means that an analysis of the mutation spectrum and intragenomic mutation rate variation between the teams can be different, more significantly than the difference in mutation rate itself: AT bias, insertion/deletion, chromosome mutation rate, and so one. I highly encourage the authors to realize a later study exploring in detail the benchmarks proposed at the end of page 30.

We agree with the reviewer that the large difference in candidate DNMs could lead to differences in the mutation spectrum. Thus, we extended our discussion on Figure 3 - supplementary fig 2, showing the spectrum variation in candidates found by different groups and a comparison of the ti/tv ratio (page 23 lines 605-608).

We thank the reviewer for this comment and invitation to pursue our exploration on the effect of different methodology not only on the rate but also on the candidate DNMs. We agree that this would be highly interesting for the community. Using a large human pedigree that is extensively studied could help us to distinguish the methodology effect. We have now added a sentence as perspective on this (page 31 lines 756-760).

Reviewer #2 (Recommendations for the authors):I had couple of suggestions:It would be very useful to introduce a site-specific error rate parameter at least for the human data.

Yes, having site-specific error rates would be very useful. In studies with many sequenced individuals, it should be possible to investigate the error rate of a site where a de novo variant is called by looking at the number of observed alternative alleles at this site in unrelated individuals. But error rates are quite low, so estimating a good independent error rate for all positions would probably require a large number of samples. Consequently, such a strategy is probably unlikely to improve de novo detection a lot except in the most extensive human studies. A second strategy is to have different error rates for different sites depending on the sequence context, but we are not aware of any good tools doing this that we could recommend in the review. We have now added a brief discussion on this point (page 35 lines 820-825).

Also it would be interesting if the authors can use some of their suggested parameters: alignment, variant calling and post-filtering on some of the available data to show how their guidelines may improve the mutation rate estimate.

We agree with the reviewer that the next step for our study would be to estimate germline mutation rate in the large available human dataset, for which an agreement of a mutation rate ~ 1.2 x 10^-8^ per site per generation at an average age of ~30 years has been found by many studies. This would help define some standard parameters for detecting germline mutation rates. Yet, we believe this is beyond the scope of our study. We have now added a sentence as a perspective on this (page 31 lines 756-760).

Reviewer #3 (Recommendations for the authors):Assessing the variability of germline mutation rates predicted from each pipeline via application to more (e.g.) pedigrees would be informative.

We agree with the reviewer that this would be interesting. We have added a sentence as a perspective for the Mutationathon to do a similar effort on a larger pedigree. We also advise always releasing sequences and vcf files in order to allow such comparison on available datasets (page 32 line 771). We also note that many published papers (especially in non-human primates) use multiple different pipelines to ensure the robustness of their results (page 31 lines 755-760).

LCRs on line 434 refer to low-complexity regions identified by Heng Li as drivers of false-positive variant predictions.

This is right. We have now changed it (page 18 line 464).

The color scheme in Figure 3B will be difficult for the color blind.

We have removed figure 3b as it was now repetitive with the upset plot.

Figure 3C is difficult to parse. Perhaps consider an "Upset" plot.

We now use an upset plot, and combine the information of panel b and c into a single figure.

Do the GQ filters discussed on lines 670-674 apply to each member of the pedigree? That is, is GQ>=20 applied to the genotype for each macaque at each site?

Yes, we try to make this more clear by distinguishing the site-specific filters and the sample-specific filters and using the box 1 page 15.

Figure 5 is very difficult to read. Please make the axis labels larger. You may be able to also make the figure more legible by packing the bars more tightly so as to waste less whitespace.

We updated Figure 5.

The manuscript is often difficult to read with all of the VCF acronyms used to describe the filtering logic applied. While decipherable for those familiar with the "joys" of VCF, you might consider a more accessible approach for a broader audience.

We agree with the reviewer. We have now added a box to explain some of the vcf jargon (page 15).